# On the Estimation Bias in Double Q-Learning

**Zhizhou Ren**[1][†]**, Guangxiang Zhu**[2]**, Hao Hu**[2]**, Beining Han**[2]**, Jianglun Chen**[2]**, Chongjie Zhang**[2]
[1]Department of Computer Science, University of Illinois at Urbana-Champaign
[2]Institute for Interdisciplinary Information Sciences, Tsinghua University
zhizhour@illinois.edu, guangxiangzhu@outlook.com
{hu-h19, hbn18, chen-jl18}@mails.tsinghua.edu.cn
chongjie@tsinghua.edu.cn

## Abstract

Double Q-learning is a classical method for reducing overestimation bias, which is caused by taking maximum estimated values in the Bellman operation. Its variants in the deep Q-learning paradigm have shown great promise in producing reliable value prediction and improving learning performance. However, as shown by prior work, double Q-learning is not fully unbiased and suffers from underestimation bias. In this paper, we show that such underestimation bias may lead to multiple non-optimal fixed points under an approximate Bellman operator. To address the concerns of converging to non-optimal stationary solutions, we propose a simple but effective approach as a partial fix for the underestimation bias in double Q-learning. This approach leverages an approximate dynamic programming to bound the target value. We extensively evaluate our proposed method in the Atari benchmark tasks and demonstrate its significant improvement over baseline algorithms.

## 1 Introduction

Value-based reinforcement learning with neural networks as function approximators has become a widely-used paradigm and shown great promise in solving complicated decision-making problems in various real-world applications, including robotics control (Lillicrap et al., 2016), molecular structure design (Zhou et al., 2019), and recommendation systems (Chen et al., 2018). Towards understanding the foundation of these successes, investigating algorithmic properties of deep-learning-based value function approximation has attracted a growth of attention in recent years (Van Hasselt et al., 2018; Fu et al., 2019; Achiam et al., 2019; Dong et al., 2020). One of the phenomena of interest is that Q-learning (Watkins, 1989) is known to suffer from overestimation issues, since it takes a maximum operator over a set of estimated action-values. Comparing with underestimated values, overestimation errors are more likely to be propagated through greedy action selections, which leads to an overestimation bias in value prediction (Thrun and Schwartz, 1993). This overoptimistic behavior of decision making has also been investigated in the literature of management science (Smith and Winkler, 2006) and economics (Thaler, 1988).

In deep Q-learning algorithms, one major source of value estimation errors comes from the optimization procedure. Although a deep neural network may have a sufficient expressiveness power to represent an accurate value function, the back-end optimization is hard to solve. As a result of computational considerations, stochastic gradient descent is almost the default choice for training deep Q-networks. As pointed out by Riedmiller (2005) and Van Hasselt et al. (2018), a mini-batch gradient update may have unpredictable effects on state-action pairs outside the training batch. The high variance of gradient estimation by such stochastic methods would lead to an unavoidable approximation error in value prediction, which cannot be eliminated by simply increasing sample size and

---

[†]Work done while Zhizhou was an undergraduate at Tsinghua University.

35th Conference on Neural Information Processing Systems (NeurIPS 2021).

network capacity. Through the maximum operator in the Q-learning paradigm, such approximation error would propagate and accumulate to form an overestimation bias. In practice, even if most benchmark environments are nearly deterministic (Brockman et al., 2016), a dramatic overestimation can be observed (Van Hasselt et al., 2016).

Double Q-learning (Van Hasselt, 2010) is a classical method to reduce the risk of overestimation, which is a specific variant of the double estimator (Stone, 1974) in the Q-learning paradigm. Instead of taking the greedy maximum values, it uses a second value function to construct an independent action-value evaluation as a cross validation. With proper assumptions, double Q-learning was proved to slightly underestimate rather than overestimate the maximum expected values (Van Hasselt, 2010). This technique has become a default implementation for stabilizing deep Q-learning algorithms (Hessel et al., 2018). In continuous control domains, a famous variant named clipped double Q-learning (Fujimoto et al., 2018) also shows great success in reducing the accumulation of errors in actor-critic methods (Haarnoja et al., 2018; Kalashnikov et al., 2018).

To understand algorithmic properties of double Q-learning and its variants, most prior work focus on the characterization of one-step estimation bias, i.e., the expected deviation from target values in a single step of Bellman operation (Lan et al., 2020; Chen et al., 2021). In this paper, we present a different perspective on how these one-step errors accumulate in stationary solutions. We first review a widely-used analytical model introduced by Thrun and Schwartz (1993) and reveal a fact that, due to the perturbation of approximation error, both double Q-learning and clipped double Q-learning have multiple approximate fixed points in this model. This result raises a concern that double Q-learning may easily get stuck in some local stationary regions and become inefficient in searching for the optimal policy. Motivated by this finding, we propose a novel value estimator, named *doubly bounded estimator*, that utilizes an abstracted dynamic programming as a lower bound estimation to rule out the potential non-optimal fixed points. The proposed method is easy to be combined with other existing techniques such as clipped double Q-learning. We extensively evaluate our approach on a variety of Atari benchmark tasks, and demonstrate significant improvement over baseline algorithms in terms of sample efficiency and convergence performance.

## 2   Background

Markov Decision Process (MDP; Bellman, 1957) is a classical framework to formalize an agent-environment interaction system which can be defined as a tuple $\mathcal{M} = \langle \mathcal{S}, \mathcal{A}, P, R, \gamma \rangle$. We use $\mathcal{S}$ and $\mathcal{A}$ to denote the state and action space, respectively. $P(s'|s,a)$ and $R(s,a)$ denote the transition and reward functions, which are initially unknown to the agent. $\gamma$ is the discount factor. The goal of reinforcement learning is to construct a policy $\pi : \mathcal{S} \to \mathcal{A}$ maximizing cumulative rewards

$$V^\pi(s) = \mathbb{E}\left[\sum_{t=0}^{\infty} \gamma^t R(s_t, \pi(s_t)) \;\middle|\; s_0 = s, s_{t+1} \sim P(\cdot|s_t, \pi(s_t))\right].$$

Another quantity of interest can be defined through the Bellman equation $Q^\pi(s,a) = R(s,a) + \gamma \mathbb{E}_{s' \sim P(\cdot|s,a)}[V^\pi(s')]$. The optimal value function $Q^*$ corresponds to the unique solution of the Bellman optimality equation, $Q^*(s,a) = R(s,a) + \gamma \mathbb{E}_{s' \sim P(\cdot|s,a)}[\max_{a' \in \mathcal{A}} Q^*(s',a')]$. Q-learning algorithms are based on the Bellman optimality operator $\mathcal{T}$ stated as follows:

$$(\mathcal{T}Q)(s,a) = R(s,a) + \gamma \mathop{\mathbb{E}}_{s' \sim P(\cdot|s,a)}\left[\max_{a' \in \mathcal{A}} Q(s',a')\right]. \tag{1}$$

By iterating this operator, value iteration is proved to converge to the optimal value function $Q^*$. To extend Q-learning methods to real-world applications, function approximation is indispensable to deal with a high-dimensional state space. Deep Q-learning (Mnih et al., 2015) considers a sample-based objective function and deploys an iterative optimization framework

$$\theta_{t+1} = \arg\min_{\theta \in \Theta} \mathop{\mathbb{E}}_{(s,a,r,s') \sim \mathcal{D}}\left[\left(r + \gamma \max_{a' \in \mathcal{A}} Q_{\theta_t}(s',a') - Q_\theta(s,a)\right)^2\right], \tag{2}$$

in which $\Theta$ denotes the parameter space of the value network, and $\theta_0 \in \Theta$ is initialized by some predetermined method. $(s,a,r,s')$ is sampled from a data distribution $\mathcal{D}$ which is changing during exploration. With infinite samples and a sufficiently rich function class, the update rule stated in

Eq. (2) is asymptotically equivalent to applying the Bellman optimality operator $\mathcal{T}$, but the underlying optimization is usually inefficient in practice. In deep Q-learning, Eq. (2) is optimized by mini-batch gradient descent and thus its value estimation suffers from unavoidable approximation errors.

# 3 On the Effects of Underestimation Bias in Double Q-Learning

In this section, we will first revisit a common analytical model used by previous work for studying estimation bias (Thrun and Schwartz, 1993; Lan et al., 2020), in which double Q-learning is known to have underestimation bias. Based on this analytical model, we show that its underestimation bias could make double Q-learning have multiple fixed-point solutions under an approximate Bellman optimality operator. This result suggests that double Q-learning may have extra non-optimal stationary solutions under the effects of the approximation error.

## 3.1 Modeling Approximation Error in Q-Learning

In Q-learning with function approximation, the ground truth Bellman optimality operator $\mathcal{T}$ is approximated by a regression problem through Bellman error minimization (see Eq. (1) and Eq. (2)), which may suffer from unavoidable approximation errors. Following Thrun and Schwartz (1993) and Lan et al. (2020), we formalize underlying approximation errors as a set of random noises $e^{(t)}(s, a)$ on the regression outcomes:

$$Q^{(t+1)}(s, a) = (\mathcal{T}Q^{(t)})(s, a) + e^{(t)}(s, a). \tag{3}$$

In this model, double Q-learning (Van Hasselt, 2010) can be modeled by two estimator instances $\{Q_i^{(t)}\}_{i \in \{1,2\}}$ with separated noise terms $\{e_i^{(t)}\}_{i \in \{1,2\}}$. For simplification, we introduce a policy function $\pi^{(t)}(s) = \arg\max_a Q_1^{(t)}(s, a)$ to override the state value function as follows:

$$V^{(t)}(s) = Q_2^{(t)}\left(s,\ \pi^{(t)}(s) = \arg\max_{a \in \mathcal{A}} Q_1^{(t)}(s, a)\right),$$

$$\forall i \in \{1, 2\}, \quad Q_i^{(t+1)}(s, a) = \underbrace{R(s, a) + \gamma \mathbb{E}_{s'}\left[V^{(t)}(s')\right]}_{\text{target value}} + \underbrace{e_i^{(t)}(s, a)}_{\text{approximation error}}. \tag{4}$$

A minor difference of Eq. (4) from the definition of double Q-learning given by Van Hasselt (2010) is the usage of a unified target value $V^{(t)}(s')$ for both two estimators. This simplification does not affect the derived implications, and is also implemented by advanced variants of double Q-learning (Fujimoto et al., 2018; Lan et al., 2020).

To establish a unified framework for analysis, we use a stochastic operator $\widetilde{\mathcal{T}}$ to denote the Q-iteration procedure $Q^{(t+1)} \leftarrow \widetilde{\mathcal{T}}Q^{(t)}$, e.g., the updating rules stated as Eq. (3) and Eq. (4). We call such an operator $\widetilde{\mathcal{T}}$ as a *stochastic Bellman operator*, since it approximates the ground truth Bellman optimality operator $\mathcal{T}$ and carries some noises due to approximation errors. Note that, as shown in Eq. (4), the target value can be constructed only using the state-value function $V^{(t)}$. We can define the stationary point of state-values $V^{(t)}$ as the fixed point of a stochastic Bellman operator $\widetilde{\mathcal{T}}$.

**Definition 1** (Approximate Fixed Points). *Let $\widetilde{\mathcal{T}}$ denote a stochastic Bellman operator, such as what are stated in Eq. (3) and Eq. (4). A state-value function $V$ is regarded as an approximate fixed point under a stochastic Bellman operator $\widetilde{\mathcal{T}}$ if it satisfies $\mathbb{E}[\widetilde{\mathcal{T}}V] = V$, where $\widetilde{\mathcal{T}}V$ denotes the output state-value function while applying the Bellman operator $\widetilde{\mathcal{T}}$ on $V$.*

**Remark.** In prior work (Thrun and Schwartz, 1993), value estimation bias is defined by expected one-step deviation with respect to the ground truth Bellman operator, i.e., $\mathbb{E}[(\widetilde{\mathcal{T}}V^{(t)})(s)] - (\mathcal{T}V^{(t)})(s)$. The approximate fixed points stated in Definition 1 characterizes the accumulation of estimation biases in stationary solutions.

In Appendix A.2, we will prove the existence of such fixed points as the following statement.

**Proposition 1.** *Assume the probability density functions of the noise terms $\{e(s, a)\}$ are continuous. The stochastic Bellman operators defined by Eq. (3) and Eq. (4) must have approximate fixed points in arbitrary MDPs.*

$R_{0,0}=1.1$ $R_{1,0}=R_{1,1}=1$

$R_{0,1}=1$

$s_0 \longrightarrow s_1$

$\epsilon = 1.0$ $\gamma = 0.99$

| $V(s_0)$ | $V(s_1)$ | $\tilde{\pi}(a_0|s_0)$ |
|---|---|---|
| 100.162 | 100.0 | 62.2% |
| 101.159 | 100.0 | 92.9% |
| 110.0 | 100.0 | 100.0% |

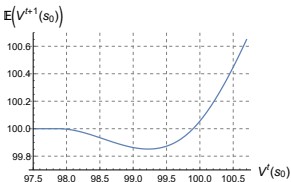

(a) A simple construction   (b) Numerical solutions of fixed points   (c) Visualizing non-monotonicity

Figure 1: (a) A simple infinite-horizon MDP where double Q-learning stated as (4) has multiple approximate fixed points. $R_{i,j}$ is a shorthand of $R(s_i, a_j)$. (b) The numerical solutions of the fixed points produced by double Q-learning in the MDP presented above. $\tilde{\pi}$ denotes the expected policy generated by the corresponding fixed point under the perturbation of noise $e(s,a)$. A formal description of $\tilde{\pi}$ refers to Definition 2 in Appendix A.3. (c) The relation between the input state-value $V^{(t)}(s_0)$ and the expected output state-value $\mathbb{E}[V^{(t+1)}(s_0)]$ generated by double Q-learning in the constructed MDP, in which we assume $V^{(t)}(s_1) = 100$.

## 3.2 Existence of Multiple Approximate Fixed Points in Double Q-Learning Algorithms

Given the definition of the approximate fixed point, a natural question is whether such kind of fixed points are unique or not. Recall that the optimal value function $Q^*$ is the unique solution of the Bellman optimality equation, which is the foundation of Q-learning algorithms. However, in this section, we will show that, under the effects of the approximation error, the approximate fixed points of double Q-learning may not be unique.

**An Illustrative Example.**   Figure 1a presents a simple MDP in which double Q-learning stated as Eq. (4) has multiple approximate fixed points. For simplicity, this MDP is set to be fully deterministic and contains only two states. All actions on state $s_1$ lead to a self-loop and produce a unit reward signal. On state $s_0$, the result of executing action $a_0$ is a self-loop with a slightly larger reward signal than choosing action $a_1$ which leads to state $s_1$. The only challenge for decision making in this MDP is to distinguish the outcomes of executing action $a_0$ and $a_1$ on state $s_0$. To make the example more accessible, we assume the approximation errors $\{e^{(t)}(s,a)\}_{t,s,a}$ are a set of independent random noises sampled from a uniform distribution $Uniform(-\epsilon, \epsilon)$. This simplification is also adopted by Thrun and Schwartz (1993) and Lan et al. (2020) in case studies. Here, we select the magnitude of noise as $\epsilon = 1.0$ and the discount factor as $\gamma = 0.99$ to balance the scale of involved amounts.

Considering to solve the equation $\mathbb{E}[\widetilde{\mathcal{T}}V] = V$ according to the definition of the approximate fixed point (see Definition 1), the numerical solutions of such fixed points are presented in Table 1b. There are three different fixed point solutions. The first thing to notice is that the optimal fixed point $V^*$ is retained in this MDP (see the last row of Table 1b), since the noise magnitude $\epsilon = 1.0$ is much smaller than the optimality gap $Q^*(s_0, a_0) - Q^*(s_0, a_1) = 10$. The other two fixed points are non-optimal and very close to $Q(s_0, a_0) \approx Q(s_0, a_1) = 100$. Intuitively, under the perturbation of approximation error, the agent cannot identify the correct maximum-value action for policy improvement in these situations, which is the cause of such non-optimal fixed points. To formalize the implications, we would present a sufficient condition for the existence of multiple extra fixed points.

**Mathematical Condition.**   Note that the definition of a stochastic Bellman operator can be decoupled to two parts: (1) Computing target values $\mathcal{T}Q^{(t)}$ according to the given MDP; (2) Perform an imprecise regression and some specific computations to obtain $Q^{(t+1)}$. The first part is defined by the MDP, and the second part is the algorithmic procedure. From this perspective, we can define the input of a learning algorithm as a set of ground truth target values $\{(\mathcal{T}Q^{(t)})(s,a)\}_{s,a}$. Based on this notation, a sufficient condition for the existence of multiple fixed points is stated as follows.

**Proposition 2.** *Let* $f_s\left(\{(\mathcal{T}Q)(s,a)\}_{a \in \mathcal{A}}\right) = \mathbb{E}[(\widetilde{\mathcal{T}}V)(s)]$ *denote the expected output value of a learning algorithm on state* $s$. *Assume* $f_s(\cdot)$ *is differentiable. If the algorithmic procedure* $f_s(\cdot)$ *satisfies Eq. (5), there exists an MDP such that it has multiple approximate fixed points.*

$$\exists s, \ \exists i, \ \exists X \in \mathbb{R}^{|\mathcal{A}|}, \quad \frac{\partial}{\partial x_i} f_s(X) > 1, \tag{5}$$

*where* $X = \{x_i\}_{i=1}^{|\mathcal{A}|}$ *denotes the input of the function* $f_s$.

The proof of Proposition 2 is deferred to Appendix A.4. This proposition suggests that, in order to determine whether a Q-learning algorithm may have multiple fixed points, we need to check whether its expected output values could change dramatically with a slight alter of inputs. Considering the constructed MDP as an example, Figure 1c visualizes the relation between the input state-value $V^{(t)}(s_0)$ and the expected output state-value $\mathbb{E}[V^{(t+1)}(s_0)]$ while assuming $V^{(t)}(s_1) = 100$ has converged to its stationary point. The minima point of the output value is located at the situation where $V^{(t)}(s_0)$ is slightly smaller than $V^{(t)}(s_1)$, since the expected policy derived by $\widetilde{\mathcal{T}}V^{(t)}$ will have a remarkable probability to choose sub-optimal actions. This local minima suffers from the most dramatic underestimation among the whole curve, and the underestimation will eventually vanish as the value of $V^{(t)}(s_0)$ increases. During this process, a large magnitude of the first-order derivative could be found to meet the condition stated in Eq. (5).

**Remark.** In Appendix A.5, we show that clipped double Q-learning, a popular variant of double Q-learning, has multiple fixed points in an MDP slightly modified from Figure 1a. Besides, the condition presented in Proposition 2 does not hold in standard Q-learning that uses a single maximum operator (see Proposition 6 in Appendix). It remains an open question whether standard Q-learning with overestimation bias has multiple fixed points.

## 3.3 Diagnosing Non-Optimal Fixed Points

In this section, we first characterize the properties of the extra non-optimal fixed points of double Q-learning in the analytical model. And then, we discuss its connections to the literature of stochastic optimization, which motivates our proposed algorithm in section 4.

**Underestimated Solutions.** The first notable thing is that, the non-optimal fixed points of double Q-learning would not overestimate the true maximum values. More specifically, every fixed-point solution could be characterized as the ground truth value of some stochastic policy as the follows:

**Proposition 3** (Fixed-Point Characterization). *Assume the noise terms $e_1$ and $e_2$ are independently generated in the double estimator stated in Eq. (4). Every approximate fixed point $V$ is equal to the ground truth value function $V^{\tilde{\pi}}$ with respect to a stochastic policy $\tilde{\pi}$.*

The proof of Proposition 3 is deferred to Appendix A.3. In addition, the corresponding stochastic policy $\tilde{\pi}$ can be interpreted as

$$\tilde{\pi}(a|s) = \mathbb{P}\left[a = \arg\max_{a' \in \mathcal{A}}\left(\underbrace{R(s,a') + \gamma\mathbb{E}_{s'}\left[V(s')\right]}_{(\mathcal{T}Q)(s,a')} + e(s,a')\right)\right],$$

which is the expected policy generated by the corresponding fixed point along with the random noise $e(s, a')$. This stochastic policy, named as *induced policy*, can provide a snapshot to infer how the agent behaves and evolves around these approximate fixed points. To deliver intuitions, we provide an analogical explanation in the context of optimization as the following arguments.

**Analogy with Saddle Points.** Taking the third column of Table 1b as an example, due to the existence of the approximation error, the induced policy $\tilde{\pi}$ suffers from a remarkable uncertainty in determining the best action on state $s_0$. Around such non-optimal fixed points, the greedy action selection may be disrupted by approximation error and deviate from the correct direction for policy improvement. These approximate fixed points are not necessary to be strongly stationary solutions but may seriously hurt the learning efficiency. If we imagine each iteration of target updating as a step of "*gradient update*" for Bellman error minimization, the non-optimal fixed points would refer to the concept of *saddle points* in the context of optimization. As stochastic gradient may be trapped in saddle points, Bellman operation with approximation error may get stuck around non-optimal approximate fixed points. Please refer to section 5.1 for a visualization of a concrete example.

**Escaping from Saddle Points.** In the literature of non-convex optimization, the most famous approach to escaping saddle points is *perturbed gradient descent* (Ge et al., 2015; Jin et al., 2017). Recall that, although gradient directions are ambiguous around saddle points, they are not strongly convergent solutions. Some specific perturbation mechanisms with certain properties could help to make the optimizer to escape non-optimal saddle points. Although these methods cannot be directly applied to double Q-learning since the Bellman operation is not an exact gradient descent, it motivates us to construct a specific perturbation as guidance. In section 4, we would introduce a perturbed target updating mechanism that uses an external value estimation to rule out non-optimal fixed points.

# 4 Doubly Bounded Q-Learning through Abstracted Dynamic Programming

As discussed in the last section, the underestimation bias of double Q-learning may lead to multiple non-optimal fixed points in the analytical model. A major source of such underestimation is the inherent approximation error caused by the imprecise optimization. Motivated by the literature of escaping saddle points, we introduce a novel method, named *Doubly Bounded Q-learning*, which integrates two different value estimators to reduce the negative effects of underestimation.

## 4.1 Algorithmic Framework

As discussed in section 3.3, the geometry property of non-optimal approximate fixed points of double Q-learning is similar to that of saddle points in the context of non-convex optimization. The theory of escaping saddle points suggests that, a well-shaped perturbation mechanism could help to remove non-optimal saddle points from the landscape of optimization (Ge et al., 2015; Jin et al., 2017). To realize this brief idea in the specific context of iterative Bellman error minimization, we propose to integrate a second value estimator using different learning paradigm as an external auxiliary signal to rule out non-optimal approximate fixed points of double Q-learning. To give an overview, we first revisit two value estimation paradigms as follows:

1. **Bootstrapping Estimator:** As the default implementation of most temporal-difference learning algorithms, the target value $y^{\text{Boots}}$ of a transition sample $(s_t, a_t, r_t, s_{t+1})$ is computed through bootstrapping the latest value function back-up $V_{\theta_{\text{target}}}$ parameterized by $\theta_{\text{target}}$ on the successor state $s_{t+1}$ as follows:

$$y^{\text{Boots}}_{\theta_{\text{target}}}(s_t, a_t) = r_t + \gamma V_{\theta_{\text{target}}}(s_{t+1}),$$

   where the computations of $V_{\theta_{\text{target}}}$ differ in different algorithms (e.g., different variants of double Q-learning).

2. **Dynamic Programming Estimator:** Another approach to estimating state-action values is applying dynamic programming in an abstracted MDP (Li et al., 2006) constructed from the collected dataset. By utilizing a state aggregation function $\phi(s)$, we could discretize a complex environment to a manageable tabular MDP. The reward and transition functions of the abstracted MDP are estimated through the collected samples in the dataset. An alternative target value $y^{\text{DP}}$ is computed as:

$$y^{\text{DP}}(s_t, a_t) = r_t + \gamma V^*_{\text{DP}}(\phi(s_{t+1})), \tag{6}$$

   where $V^*_{\text{DP}}$ corresponds to the optimal value function of the abstracted MDP.

The advantages and bottlenecks of these two types of value estimators lie in different aspects of error controlling. The generalizability of function approximators is the major strength of the *bootstrapping estimator*, but on the other hand, the hardness of the back-end optimization would cause considerable approximation error and lead to the issues discussed in section 3. The tabular representation of the *dynamic programming estimator* would not suffer from systematic approximation error during optimization, but its performance relies on the accuracy of state aggregation and the sampling error in transition estimation.

**Doubly Bounded Estimator.** To establish a trade-off between the considerations in the above two value estimators, we propose to construct an integrated estimator, named *doubly bounded estimator*, which takes the maximum values over two different basis estimation methods:

$$y^{\text{DB}}_{\theta_{\text{target}}}(s_t, a_t) = \max\left\{ y^{\text{Boots}}_{\theta_{\text{target}}}(s_t, a_t),\ y^{\text{DP}}(s_t, a_t) \right\}. \tag{7}$$

The targets values $y^{\text{DB}}_{\theta_{\text{target}}}$ would be used in training the parameterized value function $Q_\theta$ by minimizing

$$L(\theta; \theta_{\text{target}}) = \mathop{\mathbb{E}}_{(s_t, a_t)\sim D} \left( Q_\theta(s_t, a_t) - y^{\text{DB}}_{\theta_{\text{target}}}(s_t, a_t) \right)^2,$$

where $D$ denotes the experience buffer. Note that, this estimator maintains two value functions using different data structures. $Q_\theta$ is the major value function which is used to generate the behavior policy for both exploration and evaluation. $V_{\text{DP}}$ is an auxiliary value function computed by the abstracted dynamic programming, which is stored in a discrete table. The only functionality of $V_{\text{DP}}$ is computing the auxiliary target value $y^{\text{DP}}$ used in Eq. (7) during training.

**Remark.** The name "*doubly bounded*" refers to the following intuitive motivation: Assume both basis estimators, $y^{\text{Boots}}$ and $y^{\text{DP}}$, are implemented by their conservative variants and do not tend to overestimate values. The doubly bounded target value $y^{\text{DB}}(s_t, a_t)$ would become a good estimation if either of basis estimator provides an accurate value prediction on the given $(s_t, a_t)$. The outcomes of abstracted dynamic programming could help the bootstrapping estimator to escape the non-optimal fixed points of double Q-learning. The function approximator used by the bootstrapping estimator could extend the generalizability of discretization-based state aggregation. The learning procedure could make progress if either of estimators can identify the correct direction for policy improvement.

**Practical Implementation.** To make sure the dynamic programming estimator does not overestimate the true values, we implement a tabular version of batch-constrained Q-learning (BCQ; Fujimoto et al., 2019) to obtain a conservative estimation. The abstracted MDP is constructed by a simple state aggregation based on low-resolution discretization, i.e., we only aggregate states that cannot be distinguished by visual information. We follow the suggestions given by Fujimoto et al. (2019) and Liu et al. (2020) to prune the unseen state-action pairs in the abstracted MDP. The reward and transition functions of remaining state-action pairs are estimated through the average of collected samples. A detailed description is deferred to Appendix B.5.

### 4.2 Underlying Bias-Variance Trade-Off

In general, there is no existing approach can completely eliminate the estimation bias in Q-learning algorithm. Our proposed method also focuses on the underlying bias-variance trade-off.

**Provable Benefits on Variance Reduction.** The algorithmic structure of the proposed *doubly bounded estimator* could be formalized as a stochastic Bellman operator $\widetilde{\mathcal{T}}^{\text{DB}}$:

$$(\widetilde{\mathcal{T}}^{\text{DB}}V)(s) = \max\left\{(\widetilde{\mathcal{T}}^{\text{Boots}}V)(s),\ V^{\text{DP}}(s)\right\}, \tag{8}$$

where $\widetilde{\mathcal{T}}^{\text{Boots}}$ is the stochastic Bellman operator corresponding to the back-end bootstrapping estimator (e.g., Eq. (4)). $V^{\text{DP}}$ is an arbitrary deterministic value estimator such as using abstracted dynamic programming. The benefits on variance reduction can be characterized as the following proposition.

**Proposition 4.** *Given an arbitrary stochastic operator $\widetilde{\mathcal{T}}^{Boots}$ and a deterministic estimator $V^{DP}$,*

$$\forall V,\ \forall s \in \mathcal{S}, \quad Var[(\widetilde{\mathcal{T}}^{DB}V)(s)] \leq Var[(\widetilde{\mathcal{T}}^{Boots}V)(s)],$$

*where $(\widetilde{\mathcal{T}}^{DB}V)(s)$ is defined as Eq. (8).*

The proof of Proposition 4 is deferred to Appendix A.6. The intuition behind this statement is that, with a deterministic lower bound cut-off, the variance of the outcome target values would be reduced, which may contribute to improve the stability of training.

**Trade-Off between Different Biases.** In general, the proposed *doubly bounded estimator* does not have a rigorous guarantee for bias reduction, since the behavior of abstracted dynamic programming depends on the properties of the tested environments and the accuracy of state aggregation. In the most unfavorable case, if the dynamic programming component carries a large magnitude of error, the lower bounded objective would propagate high-value errors to increase the risk of overestimation. To address these concerns, we propose to implement a conservative approximate dynamic programming as discussed in the previous section. The asymptotic behavior of batch-constrained Q-learning does not tend to overestimate extrapolated values (Liu et al., 2020). The major risk of the dynamic programming module is induced by the state aggregation, which refers to a classical problem (Li et al., 2006). The experimental analysis in section 5.2 demonstrates that, the error carried by abstracted dynamic programming is acceptable, and it definitely works well in most benchmark tasks.

## 5 Experiments

In this section, we conduct experiments to demonstrate the effectiveness of our proposed method[1]. We first perform our method on the two-state MDP example discussed in previous sections to visualize its algorithmic effects. And then, we set up a performance comparison on Atari benchmark with several baseline algorithms. A detailed description of experiment settings is deferred to Appendix B.

---

[1]Our code is available at `https://github.com/Stilwell-Git/Doubly-Bounded-Q-Learning`.

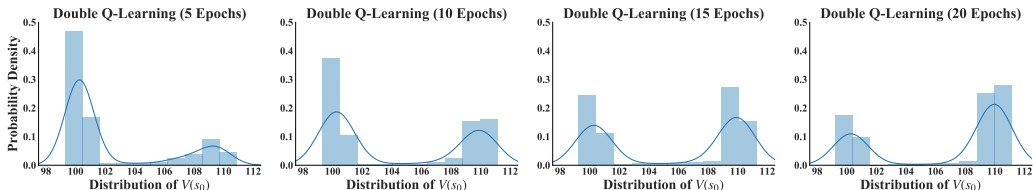

Figure 2: Visualization of the probability density of $V(s_0)$ learned by double Q-learning. In this section, all plotted distributions are estimated by $10^3$ runs with random seeds. We utilize `seaborn.distplot` package to plot the kernel density estimation curves and histogram bins.

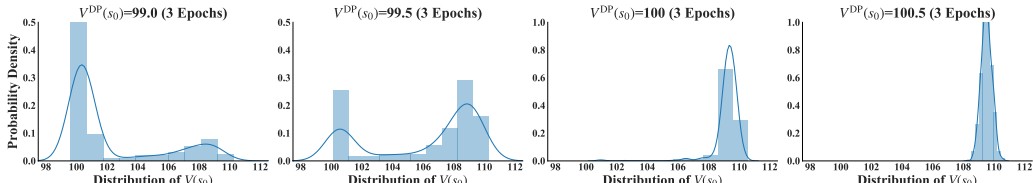

Figure 3: Visualization of the probability density of $V(s_0)$ learned by doubly bounded Q-learning with an imprecise dynamic programming planner. $V^{\mathrm{DP}}(s_0)$ denotes the value estimation given by dynamic programming, which serves as a lower bound in our method.

## 5.1 Tabular Experiments on a Two-State MDP

To investigate whether the empirical behavior of our proposed method matches it design purpose, we compare the behaviors of double Q-learning and our method on the two-state MDP example presented in Figure 1a. In this MDP, the non-optimal fixed points would get stuck in $V(s_0) \approx 100$ and the optimal solution has $V^*(s_0) = 110$. In this tabular experiment, we implement double Q-learning and our algorithm with table-based Q-value functions. The Q-values are updated by iteratively applying Bellman operators as what are presented in Eq. (4) and Eq. (8). To approximate practical scenarios, we simulate the approximation error by entity-wise Gaussian noises $\mathcal{N}(0, 0.5)$. Since the stochastic process induced by such noises suffers from high variance, we perform soft update $Q^{(t+1)} = (1 - \alpha)Q^{(t)} + \alpha(\widetilde{\mathcal{T}}Q^{(t)})$ to make the visualization clear, in which $\alpha$ refers to learning rate in practice. We consider $\alpha = 10^{-2}$ for all experiments presented in this section. In this setting, we denote one epoch as $\frac{1}{\alpha(1-\gamma)} = 10^4$ iterations. A detailed description for the tabular implementation is deferred to Appendix B.2.

We investigate the performance of our method with an imprecise dynamic programming module, in which we only apply lower bound for state $s_0$ with values $V^{\mathrm{DP}}(s_0) \in \{99.0, 99.5, 100.0, 100.5\}$. The experiment results presented in Figure 2 and Figure 3 support our claims in previous sections:

1. The properties of non-optimal fixed points are similar to that of saddle points. These extra fixed points are relatively stationary region but not truly static. A series of lucky noises can help the agent to escape from non-optimal stationary regions, but this procedure may take lots of iterations. As shown in Figure 2, double Q-learning may get stuck in non-optimal solutions and it can escape these non-optimal regions by a really slow rate. After 20 epochs (i.e., $2 \cdot 10^5$ iterations), there are nearly a third of runs cannot find the optimal solution.

2. The design of our doubly bounded estimator is similar to a perturbation on value learning. As shown in Figure 3, when the estimation given by dynamic programming is slightly higher than the non-optimal fixed points, such as $V_{\mathrm{DP}}(s_0) = 100.5$, it is sufficient to help the agent escape from non-optimal stationary solutions. A tricky observation is that $V_{\mathrm{DP}}(s_0) = 100$ also seems to work. It is because cutting-off a zero-mean noise would lead to a slight overestimation, which makes the actual estimated value of $V^{\mathrm{DP}}(s_0)$ to be a larger value.

## 5.2 Performance Comparison on Atari Benchmark

To demonstrate the superiority of our proposed method, *Doubly Bounded Q-Learning through Abstracted Dynamic Programming* (DB-ADP), we compare with six variants of deep Q-networks as baselines, including DQN (Mnih et al., 2015), double DQN (DDQN; Van Hasselt et al., 2016),

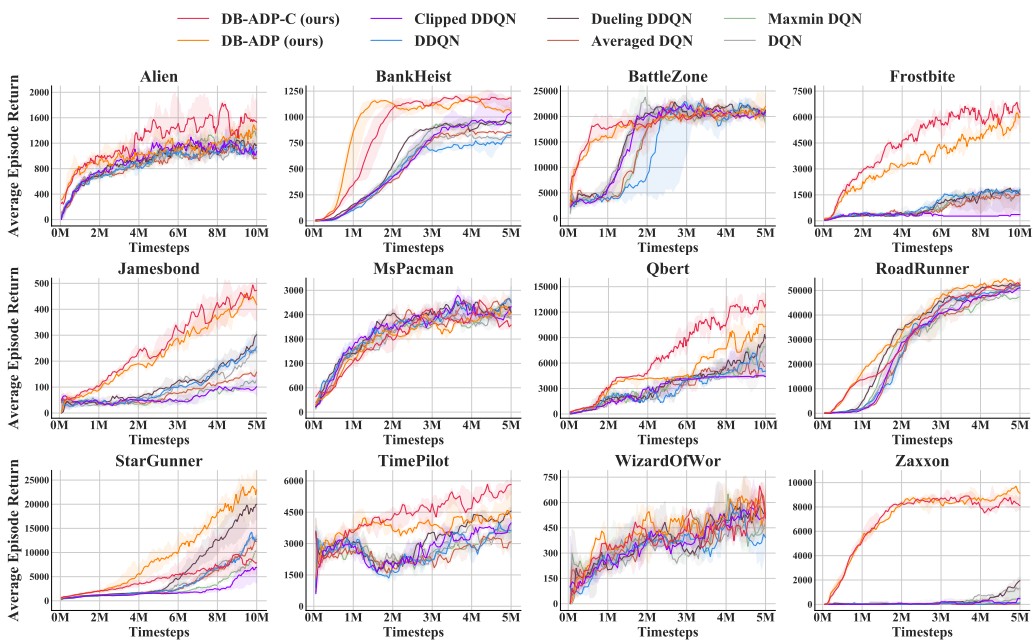

Figure 4: Learning curves on a suite of Atari benchmark tasks. DB-ADP and DB-ADP-C refer to our proposed approach built upon double Q-learning and clipped double Q-learning, respectively.

dueling DDQN (Wang et al., 2016), averaged DQN (Anschel et al., 2017), maxmin DQN (Lan et al., 2020), and clipped double DQN adapted from Fujimoto et al. (2018). Our proposed doubly bounded target estimation $y^{\mathrm{DB}}$ is built upon two types of bootstrapping estimators that have clear incentive of underestimation, i.e., double Q-learning and clipped double Q-learning. We denote these two variants as DB-ADP-C and DB-ADP according to our proposed method with or without using clipped double Q-learning.

As shown in Figure 4, the proposed doubly bounded estimator has great promise in bootstrapping the performance of double Q-learning algorithms. The improvement can be observed both in terms of sample efficiency and final performance. Another notable observation is that, although clipped double Q-learning can hardly improve the performance upon Double DQN, it can significantly improve the performance through our proposed approach in most environments (i.e., DB-ADP-C vs. DB-ADP in Figure 4). This improvement should be credit to the conservative property of clipped double Q-learning (Fujimoto et al., 2019) that may reduce the propagation of the errors carried by abstracted dynamic programming.

## 5.3 Variance Reduction on Target Values

To support the theoretical claims in Proposition 4, we conduct an experiment to demonstrate the ability of doubly bounded estimator on variance reduction. We evaluate the standard deviation of the target values with respect to training networks using different sequences of training batches. Table 5a presents the evaluation results on our proposed methods and baseline algorithms. The †-version corresponds to an ablation study, where we train the network using our proposed approach but evaluate the target values computed by bootstrapping estimators, i.e., using the target value formula of double DQN or clipped double DQN. As shown in Table 5a, the standard deviation of target values is significantly reduced by our approaches, which matches our theoretical analysis in Proposition 4. It demonstrates a strength of our approach in improving training stability. A detailed description of the evaluation metric is deferred to Appendix B.

## 5.4 An Ablation Study on the Dynamic Programming Module

To support the claim that the dynamic programming estimator is an auxiliary module to improving the strength of double Q-learning, we conduct an ablation study to investigate the individual performance of dynamic programming. Formally, we exclude Bellman error minimization from the training

| TASK NAME | DB-ADP-C | DB-ADP-C† | CDDQN |
|---|---|---|---|
| ALIEN | **0.006** | 0.008 | 0.010 |
| BANKHEIST | **0.009** | 0.010 | 0.010 |
| QBERT | **0.008** | 0.010 | 0.011 |

| TASK NAME | DB-ADP | DB-ADP† | DDQN |
|---|---|---|---|
| ALIEN | 0.008 | 0.009 | 0.012 |
| BANKHEIST | **0.009** | 0.011 | 0.013 |
| QBERT | 0.009 | 0.011 | 0.012 |

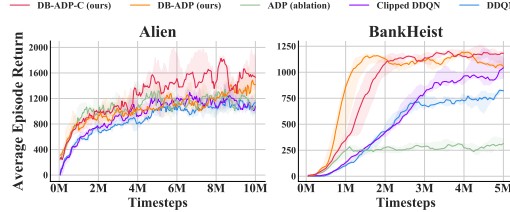

(a) Variance reduction on target values      (b) Ablation study on dynamic programming

Figure 5: (a) Evaluating the standard deviation of target values w.r.t. different training batches. The presented amounts are normalized by the value scale of corresponding runs. "†" refers to ablation studies. (b) An ablation study on the individual performance of the dynamic programming module.

procedure and directly optimize the following objective to distill the results of dynamic programming into a generalizable parametric agent:

$$L^{\mathrm{ADP}}(\theta) = \mathop{\mathbb{E}}_{(s_t, a_t) \sim D} \left[ \left( Q_\theta(s_t, a_t) - y^{\mathrm{DP}}(s_t, a_t) \right)^2 \right],$$

where $y^{\mathrm{DP}}(s_t, a_t)$ denotes the target value directly by dynamic programming. As shown in Figure 5b, without integrating with the bootstrapping estimator, the abstracted dynamic programming itself cannot outperform deep Q-learning algorithms. It remarks that, in our proposed framework, two basis estimators are supplementary to each other.

# 6 Related Work

Correcting the estimation bias of Q-learning is a long-lasting problem which induces a series of approaches (Lee et al., 2013; D'Eramo et al., 2016; Chen, 2020; Zhang and Huang, 2020), especially following the methodology of double Q-learning (Zhang et al., 2017; Zhu and Rigotti, 2021). The most representative algorithm, clipped double Q-learning (Fujimoto et al., 2018), has become the default implementation of most advanced actor-critic algorithms (Haarnoja et al., 2018). Based on clipped double Q-learning, several methods have been investigated to reduce the its underestimation and achieve promising performance (Ciosek et al., 2019; Li and Hou, 2019). Other recent advances usually focus on using ensemble methods to further reduce the error magnitude (Lan et al., 2020; Kuznetsov et al., 2020; Chen et al., 2021). Statistical analysis of double Q-learning is also an active area (Weng et al., 2020; Xiong et al., 2020) that deserves future studies.

Besides the variants of double Q-learning, using the softmax operator in Bellman operations is another effective approach to reduce the effects of approximation error (Fox et al., 2016; Asadi and Littman, 2017; Song et al., 2019; Kim et al., 2019; Pan et al., 2019). The characteristic of our approach is the usage of an approximate dynamic programming. Our analysis would provide a theoretical support for memory-based approaches, such as episodic control (Blundell et al., 2016; Pritzel et al., 2017; Lin et al., 2018; Zhu et al., 2020; Hu et al., 2021), which are usually designed for near-deterministic environments. Instead of using an explicit planner, Fujita et al. (2020) adopts the trajectory return as a lower bound for value estimation. This simple technique also shows promise in improving the efficiency of continuous control with clipped double Q-learning.

# 7 Conclusion

In this paper, we reveal an interesting fact that, under the effects of approximation error, double Q-learning may have multiple non-optimal fixed points. The main cause of such non-optimal fixed points is the underestimation bias of double Q-learning. Regarding this issue, we provide some analysis to characterize what kind of Bellman operators may suffer from the same problem, and how the agent may behave around these fixed points. To address the potential risk of converging to non-optimal solutions, we propose doubly bounded Q-learning to reduce the underestimation in double Q-learning. The main idea of this approach is to leverage an abstracted dynamic programming as a second value estimator to rule out non-optimal fixed points. The experiments show that the proposed method has shown great promise in improving both sample efficiency and convergence performance, which achieves a significant improvement over baselines algorithms.

## Acknowledgments and Disclosure of Funding

The authors would like to thank Kefan Dong for insightful discussions. This work is supported in part by Science and Technology Innovation 2030 – "New Generation Artificial Intelligence" Major Project (No. 2018AAA0100904), a grant from the Institute of Guo Qiang, Tsinghua University, and a grant from Turing AI Institute of Nanjing.

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
