## A Omitted Statements and Proofs

### A.1 The Relation between Estimation Bias and Approximate Fixed Points

An intuitive characterization of such fixed point solutions is considering one-step estimation bias with respect to the maximum expected value, which is defined as

$$\mathcal{E}(\widetilde{\mathcal{T}}, V, s) = \mathbb{E}[(\widetilde{\mathcal{T}}V)(s)] - (\mathcal{T}V)(s), \tag{9}$$

where $(\mathcal{T}V)(s)$ corresponds to the precise state value after applying the ground truth Bellman operation. The amount of estimation bias $\mathcal{E}$ characterizes the deviation from the standard Bellman operator $\mathcal{T}$, which can be regarded as imaginary rewards in fixed point solutions.

Every approximate fixed point solution under a stochastic Bellman operator can be characterized as the optimal value function in a modified MDP where only the reward function is changed.

**Proposition 5.** *Let $\widetilde{V}$ denote an approximation fixed point under a stochastic Bellman operator $\widetilde{\mathcal{T}}$. Define a modified MDP $\widetilde{\mathcal{M}} = \langle \mathcal{S}, \mathcal{A}, P, R + \widetilde{R}, \gamma \rangle$ based on $\mathcal{M}$, where the extra reward term is defined as*

$$\widetilde{R}(s, a) = \mathcal{E}(\widetilde{\mathcal{T}}, \widetilde{V}, s) = \mathbb{E}[(\widetilde{\mathcal{T}}\widetilde{V})(s)] - (\mathcal{T}\widetilde{V})(s),$$

*where $\mathcal{E}$ is the one-step estimation bias defined in Eq. (9). Then $\widetilde{V}$ is the optimal state-value function of the modified MDP $\widetilde{\mathcal{M}}$.*

*Proof.* Define a value function $\widetilde{Q}$ based on $\widetilde{V}$, $\forall (s, a) \in \mathcal{S} \times \mathcal{A}$,

$$\widetilde{Q}(s, a) = R(s, a) + \widetilde{R}(s, a) + \mathop{\mathbb{E}}_{s' \sim P(\cdot|s,a)}[\widetilde{V}(s')].$$

We can verify $\widetilde{Q}$ is consistent with $\widetilde{V}$, $\forall s \in \mathcal{S}$,

$$\begin{aligned}
\widetilde{V}(s) &= \mathbb{E}[(\widetilde{\mathcal{T}}\widetilde{V})(s)] \\
&= \mathbb{E}[(\widetilde{\mathcal{T}}\widetilde{V})(s)] - (\mathcal{T}\widetilde{V})(s) + \max_{a \in \mathcal{A}} \left( R(s, a) + \gamma \mathop{\mathbb{E}}_{s' \sim P(\cdot|s,a)}[\widetilde{V}(s')] \right) \\
&= \max_{a \in \mathcal{A}} \left( R(s, a) + \widetilde{R}(s, a) + \gamma \mathop{\mathbb{E}}_{s' \sim P(\cdot|s,a)}[\widetilde{V}(s')] \right) \\
&= \max_{a \in \mathcal{A}} \widetilde{Q}(s, a).
\end{aligned}$$

Let $\mathcal{T}_{\widetilde{\mathcal{M}}}$ denote the Bellman operator of $\widetilde{\mathcal{M}}$. We can verify $\widetilde{Q}$ satisfies Bellman optimality equation to prove the given statement, $\forall (s, a) \in \mathcal{S} \times \mathcal{A}$,

$$\begin{aligned}
(\mathcal{T}_{\widetilde{\mathcal{M}}}\widetilde{Q})(s, a) &= R(s, a) + \widetilde{R}(s, a) + \gamma \mathop{\mathbb{E}}_{s' \sim P(\cdot|s,a)} \left[ \max_{a' \in \mathcal{A}} \widetilde{Q}(s', a') \right] \\
&= R(s, a) + \widetilde{R}(s, a) + \gamma \mathop{\mathbb{E}}_{s' \sim P(\cdot|s,a)}[\widetilde{V}(s')] \\
&= \widetilde{Q}(s, a).
\end{aligned}$$

Thus we can see $\widetilde{V}$ is the solution of Bellman optimality equation in $\widetilde{\mathcal{M}}$. $\qquad\square$

### A.2 The Existence of Approximate Fixed Points

The key technique for proving the existence of approximate fixed points is Brouwer's fixed point theorem.

**Lemma 1.** *Let $B = [-L, -L]^d$ denote a d-dimensional bounding box. For any continuous function $f : B \to B$, there exists a fixed point $x$ such that $f(x) = x \in B$.*

*Proof.* It refers to a special case of Brouwer's fixed point theorem (Brouwer, 1911). $\qquad\square$

**Lemma 2.** *Let $\widetilde{\mathcal{T}}$ denote the stochastic Bellman operator defined by Eq. (3). There exists a real range $L$, $\forall V \in [L, -L]^{|\mathcal{S}|}$, $\mathbb{E}[\widetilde{\mathcal{T}}V] \in [L, -L]^{|\mathcal{S}|}$.*

*Proof.* Let $R_{\max}$ denote the range of the reward function for MDP $\mathcal{M}$. Let $R_e$ denote the range of the noisy term. Formally,

$$R_{\max} = \max_{(s,a)\in\mathcal{S}\times\mathcal{A}} |R(s,a)|,$$

$$R_e = \max_{s\in\mathcal{S}} \mathbb{E}\left[\max_{a\in\mathcal{A}} |e(s,a)|\right].$$

Note that the $L_\infty$-norm of state value functions satisfies $\forall V \in \mathbb{R}^{|\mathcal{S}|}$,

$$\|\mathbb{E}[\widetilde{\mathcal{T}}V]\|_\infty \leq R_{\max} + R_e + \gamma\|V\|_\infty.$$

We can construct the range $L = (R_{\max} + R_e)/(1-\gamma)$ to prove the given statement. $\qquad\square$

**Lemma 3.** *Let $\widetilde{\mathcal{T}}$ denote the stochastic Bellman operator defined by Eq. (4). There exists a real range $L$, $\forall V \in [L, -L]^{|\mathcal{S}|}$, $\mathbb{E}[\widetilde{\mathcal{T}}V] \in [L, -L]^{|\mathcal{S}|}$.*

*Proof.* Let $R_{\max}$ denote the range of the reward function for MDP $\mathcal{M}$. Formally,

$$R_{\max} = \max_{(s,a)\in\mathcal{S}\times\mathcal{A}} |R(s,a)|.$$

Note that the $L_\infty$-norm of state value functions satisfies $\forall V \in \mathbb{R}^{|\mathcal{S}|}$,

$$\|\mathbb{E}[\widetilde{\mathcal{T}}V]\|_\infty \leq R_{\max} + \gamma\|V\|_\infty.$$

We can construct the range $L = R_{\max}/(1-\gamma)$ to prove the given statement. $\qquad\square$

**Proposition 1.** *Assume the probability density functions of the noise terms $\{e(s,a)\}$ are continuous. The stochastic Bellman operators defined by Eq. (3) and Eq. (4) must have approximate fixed points in arbitrary MDPs.*

*Proof.* Let $f(V) = \mathbb{E}[\widetilde{\mathcal{T}}V]$ denote the expected return of a stochastic Bellman operation. This function is continuous because all involved formulas only contain elementary functions. The given statement is proved by combining Lemma 1, 2, and 3. $\qquad\square$

### A.3   The Induced Policy of Double Q-Learning

**Definition 2** (Induced Policy). *Given a target state-value function $V$, its induced policy $\tilde{\pi}$ is defined as a stochastic action selection according to the value estimation produced by a stochastic Bellman operation $\tilde{\pi}(a|s) =$*

$$\mathbb{P}\left[a = \arg\max_{a'\in\mathcal{A}}\left(\underbrace{R(s,a') + \gamma\mathbb{E}_{s'}\left[V(s')\right]}_{(\mathcal{T}Q)(s,a')} + e_1(s,a')\right)\right],$$

*where $\{e_1(s,a)\}_{s,a}$ are drawing from the same noise distribution as what is used by double Q-learning stated in Eq. (4).*

**Proposition 3** (Fixed-Point Characterization). *Assume the noise terms $e_1$ and $e_2$ are independently generated in the double estimator stated in Eq. (4). Every approximate fixed point $V$ is equal to the ground truth value function $V^{\tilde{\pi}}$ with respect to a stochastic policy $\tilde{\pi}$.*

*Proof.* Let $V$ denote an approximate fixed point under the stochastic Bellman operator $\widetilde{\mathcal{T}}$ defined by Eq. (4). By plugging the definition of the induced policy into the stochastic operator of double Q-learning, we can get

$$V(s) = \mathbb{E}[\widetilde{\mathcal{T}}V(s)]$$
$$= \mathbb{E}\left[(\widetilde{\mathcal{T}}Q_2)\left(s,\ \arg\max_{a\in\mathcal{A}}(\widetilde{\mathcal{T}}Q_1)(s,a)\right)\right]$$

$$= \mathbb{E}\left[ (\widetilde{\mathcal{T}}Q_2)\left(s, \ \arg\max_{a \in \mathcal{A}} ((\mathcal{T}Q_1)(s,a) + e_1(s,a)) \right) \right]$$

$$= \mathop{\mathbb{E}}_{a \sim \widetilde{\pi}(\cdot|s)} \left[ (\widetilde{\mathcal{T}}Q_2)(s,a) \right]$$

$$= \mathop{\mathbb{E}}_{a \sim \widetilde{\pi}(\cdot|s)} \left[ R(s,a) + \gamma \mathop{\mathbb{E}}_{s' \sim P(\cdot|s,a)} V(s') + e_2(s,a) \right]$$

$$= \mathop{\mathbb{E}}_{a \sim \widetilde{\pi}(\cdot|s)} \left[ R(s,a) + \gamma \mathop{\mathbb{E}}_{s' \sim P(\cdot|s,a)} V(s') \right],$$

which matches the Bellman expectation equation. $\qquad\square$

As shown by this proposition, the estimated value of a non-optimal fixed point is corresponding to the value of a stochastic policy, which revisits the incentive of double Q-learning to underestimate true maximum values.

### A.4 A Sufficient Condition for Multiple Fixed Points

**Proposition 2.** *Let $f_s \left( \{ (\mathcal{T}Q)(s,a) \}_{a \in \mathcal{A}} \right) = \mathbb{E}[(\widetilde{\mathcal{T}}V)(s)]$ denote the expected output value of a learning algorithm on state $s$. Assume $f_s(\cdot)$ is differentiable. If the algorithmic procedure $f_s(\cdot)$ satisfies Eq. (5), there exists an MDP such that it has multiple approximate fixed points.*

$$\exists s, \ \exists i, \ \exists X \in \mathbb{R}^{|\mathcal{A}|}, \quad \frac{\partial}{\partial x_i} f_s(X) > 1, \tag{5}$$

*where $X = \{x_i\}_{i=1}^{|\mathcal{A}|}$ denotes the input of the function $f_s$.*

*Proof.* Suppose $f_s$ is a function satisfying the given condition.

Let $x_i = \bar{x}$ and $X$ denote the corresponding point satisfying Eq. (5).

Let $g(x)$ denote the value of $f_s$ while only changing the input value of $x_i$ to $x$. Note that, according to Eq. (5), we have $g'(\bar{x}) > 1$.

Since $f_s$ is differentiable, we can find a small region $\bar{x}_L < \bar{x} < \bar{x}_R$ around $\bar{x}$ such that $\forall x \in [\bar{x}_L, \bar{x}_R]$, $g'(x) > 1$. And then, we have $g(\bar{x}_R) - g(\bar{x}_L) > \bar{x}_R - \bar{x}_L$.

Consider to construct an MDP with only one state (see Figure 6a as an example). We can use the action corresponding to $x_i$ to construct a self-loop transition with reward $r$. All other actions lead to a termination signal and an immediate reward, where the immediate rewards correspond to other components of $X$. By setting the discount factor as $\gamma = \frac{\bar{x}_R - \bar{x}_L}{g(\bar{x}_R) - g(\bar{x}_L)} < 1$ and the reward as $r = \bar{x}_L - \gamma g(\bar{x}_L) = \bar{x}_R - \gamma g(\bar{x}_R)$, we can find both $\bar{x}_L$ and $\bar{x}_R$ are solutions of the equation $x = r + \gamma g(x)$, in which $g(\bar{x}_L)$ and $g(\bar{x}_R)$ correspond to two fixed points of the constructed MDP. $\qquad\square$

**Proposition 6.** *Vanilla Q-learning does not satisfy the condition stated in Eq. (5) in any MDPs.*

*Proof.* In vanilla Q-learning, the expected state-value after one iteration of updates is

$$\mathbb{E}[V^{(t+1)}(s)] = \mathbb{E}\left[ \max_{a \in \mathcal{A}} Q^{(t+1)}(s,a) \right]$$

$$= \mathbb{E}\left[ \max_{a \in \mathcal{A}} \left( (\mathcal{T}Q^{(t)})(s,a) + e^{(t)}(s,a) \right) \right]$$

$$= \int_{w \in \mathbb{R}^{|\mathcal{A}|}} \mathbb{P}[e^{(t)} = w] \left( \max_{a \in \mathcal{A}} \left( (\mathcal{T}Q^{(t)})(s,a) + w(s,a) \right) \right) \mathrm{d}w.$$

Denote

$$f(\mathcal{T}Q^{(t)}, w) = \max_{a \in \mathcal{A}} \left( (\mathcal{T}Q^{(t)})(s,a) + w(s,a) \right).$$

Note that the value of $f(\mathcal{T}Q^{(t)}, w)$ is 1-Lipschitz w.r.t. each entry of $\mathcal{T}Q^{(t)}$. Thus we have $\mathbb{E}[V^{(t+1)}(s)]$ is also 1-Lipschitz w.r.t. each entry of $\mathcal{T}Q^{(t)}$. The condition stated in Eq. (5) cannot hold in any MDPs. $\qquad\square$

## A.5 A Bad Case for Clipped Double Q-Learning

The stochastic Bellman operator corresponding to clipped double Q-learning is stated as follows.

$$\forall i \in \{1,2\}, \quad Q_i^{(t+1)}(s,a) = R(s,a) + \gamma \mathop{\mathbb{E}}_{s' \sim P(\cdot|s,a)} \left[ V^{(t)}(s') \right] + e_i^{(t)}(s,a),$$

$$V^{(t)}(s) = \min_{i \in \{1,2\}} Q_i^{(t)} \left( s, \arg\max_{a \in \mathcal{A}} Q_1^{(t)}(s,a) \right). \tag{10}$$

An MDP where clipped double Q-learning has multiple fixed points is illustrated as Figure 6.

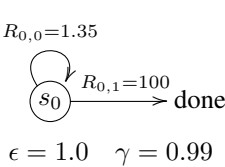

$R_{0,0} = 1.35$

$R_{0,1} = 100$ $s_0 \longrightarrow$ done

$\epsilon = 1.0 \quad \gamma = 0.99$

| $V(s_0)$ | $\tilde{\pi}(a_0|s_0)$ |
|----------|------------------------|
| 100.491  | 83.1%                  |
| 101.833  | 100.0%                 |
| 101.919  | 100.0%                 |

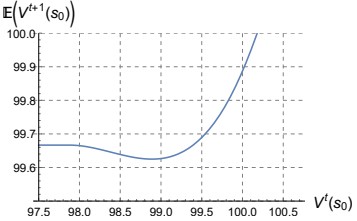

(a) A simple construction     (b) Numerical solutions of fixed points     (c) Visualizing non-monotonicity

Figure 6: (a) A simple MDP where clipped double Q-learning stated as Eq. (10) has multiple approximate fixed points. $R_{i,j}$ is a shorthand of $R(s_i, a_j)$. (b) The numerical solutions of the fixed points produced by clipped double Q-learning in the MDP presented above. (c) The relation between the input state-value $V^{(t)}(s_0)$ and the expected output state-value $\mathbb{E}[V^{(t+1)}(s_0)]$ generated by clipped double Q-learning in the constructed MDP.

## A.6 Provable Benefits on Variance Reduction

**Lemma 4.** *Let $x$ denote a random variable. $y$ denotes a constant satisfying $y \leq \mathbb{E}[x]$. Then, Var$[\max\{x,y\}] \leq$ Var$[x]$.*

*Proof.* Let $\mu = \mathbb{E}[x]$ denote the mean of random variable $x$. Consider

$$\text{Var}[x] = \int_{-\infty}^{\infty} \mathbb{P}[x=t](t-\mu)^2 \mathrm{d}t$$

$$= \int_{-\infty}^{\mu} \mathbb{P}[x=t](t-\mu)^2 \mathrm{d}t + \int_{\mu}^{\infty} \mathbb{P}[x=t](t-\mu)^2 \mathrm{d}t$$

$$\geq \int_{-\infty}^{\mu} \mathbb{P}[x=t](\mu - \max\{t,y\})^2 \mathrm{d}t + \int_{\mu}^{\infty} \mathbb{P}[x=t](t-\mu)^2 \mathrm{d}t$$

$$= \int_{-\infty}^{\infty} \mathbb{P}[x=t](\mu - \max\{t,y\})^2 \mathrm{d}t$$

$$\geq \int_{-\infty}^{\infty} \mathbb{P}[x=t](\mathbb{E}[\max\{x,y\}] - \max\{t,y\})^2 \mathrm{d}t \tag{11}$$

$$= \text{Var}[\max\{x,y\}],$$

where Eq. (11) holds since the true average point $\mathbb{E}[\max\{x,y\}]$ leads to the minimization of the variance formula. $\square$

**Proposition 4.** *Given an arbitrary stochastic operator $\widetilde{\mathcal{T}}^{Boots}$ and a deterministic estimator $V^{DP}$,*

$$\forall V, \forall s \in \mathcal{S}, \quad Var[(\widetilde{\mathcal{T}}^{DB}V)(s)] \leq Var[(\widetilde{\mathcal{T}}^{Boots}V)(s)],$$

*where $(\widetilde{\mathcal{T}}^{DB}V)(s)$ is defined as Eq. (8).*

*Proof.* When $V^{DP}(s)$ is larger than all possible output values of $(\widetilde{\mathcal{T}}^{Boots}V)(s)$, the given statement directly holds, since $\text{Var}\left[\max\left\{(\widetilde{\mathcal{T}}^{Boots}V)(s), V^{DP}(s)\right\}\right]$ would be equal to zero.

Otherwise, when $\mathbb{E}[V^{\text{Boots}}(s)]$ is smaller than $V^{\text{DP}}(s)$, we can first apply a lower bound cut-off by value $y = \mathbb{E}[V^{\text{Boots}}(s)] < V^{\text{DP}}(s)$, which gives $\text{Var}[\max\{y, V^{\text{Boots}}(s)\}] \leq \text{Var}[V^{\text{Boots}}(s)]$ and $\mathbb{E}[\max\{y, V^{\text{Boots}}(s)\}] > \mathbb{E}[V^{\text{Boots}}(s)]$. By repeating this procedure several times, we can finally get $V^{\text{DP}}(s) \leq \mathbb{E}[\max\{y, V^{\text{Boots}}(s)\}]$ and close the proof. $\qquad\square$

# B   Experiment Settings and Implementation Details

## B.1   Evaluation Settings

**Probability Density of $V(s_0)$ on Two-State MDP.**   All plotted distributions are estimated by $10^3$ runs with random seeds, i.e., suffering from different noises in approximate Bellman operations. When applying doubly bounded Q-learning, we only set lower bound for $V(s_0)$ and do nothing for $V(s_1)$. The initial Q-values are set to 100.0 as default. The main purpose of setting this initial value is avoiding the trivial case where doubly bounded Q-learning learns $V(s_0)$ faster than $V(s_1)$ so that does not get trapped in non-optimal fixed points at all. We utilize `seaborn.distplot` package to plot the kernel density estimation curves and histogram bins.

**Cumulative Rewards on Atari Games.**   All curves presented in this paper are plotted from the median performance of 5 runs with random initialization. To make the comparison more clear, the curves are smoothed by averaging 10 most recent evaluation points. The shaded region indicates 60% population around median. The evaluation is processed in every 50000 timesteps. Every evaluation point is averaged from 5 trajectories. Following Castro et al. (2018), the evaluated policy is combined with a 0.1% random execution.

**Standard Deviation of Target Values.**   The evaluation of target value standard deviations contains the following steps:

1. Every entry of the table presents the median performance of 5 runs with random network initialization.

2. For each run, we first perform $10^6$ regular training steps to collect an experience buffer and obtain a basis value function.

3. We perform a target update operation, i.e., we use the basis value function to construct frozen target values. And then we train the current values for 8000 batches as the default training configuration to make sure the current value nearly fit the target.

4. We sample a batch of transitions from the replay buffer as the testing set. We focus on the standard deviations of value predictions on this testing set.

5. And then we collect 8000 checkpoints. These checkpoints are collected by training 8000 randomly sampled batches successively, i.e., we collect one checkpoint after perform each batch updating.

6. For each transition in the testing set, we compute the standard deviation over all checkpoints. We average the standard deviation evaluation of each single transition as the evaluation of the given algorithm.

## B.2   Tabular Implementation of Doubly Bounded Q-Learning

---

**Algorithm 1** Tabular Simulation of Doubly Bounded Q-Learning

---

1: Initialize $V^{(0)}$
2: **for** $t = 1, 2, \cdots, T - 1$ **do**
3:     **for** $(s, a) \in \mathcal{S} \times \mathcal{A}$ **do**
4:         $Q^{(t)}(s, a) \leftarrow \mathbb{E}[r + \gamma V^{(t)}(s')]$
5:     $Q_1 \leftarrow Q^{(t)} + \text{noise}$
6:     $Q_2 \leftarrow Q^{(t)} + \text{noise}$
7:     **for** $s \in \mathcal{S}$ **do**
8:         $V^{(t+1)}(s) \leftarrow (1 - \alpha)V^{(t)}(s) + \alpha \max\{Q_2(s, \arg\max_a Q_1(s, a)), V_{\text{DP}}(s)\}$
9: **return** $V^{(T)}$

---

## B.3 DQN-Based Implementation of Double Bounded Q-Learning

---

**Algorithm 2** DQN-Based Implementation of Doubly Bounded Q-Learning

---

1: Initialize $\theta$, $\theta^{\text{target}}$
2: $\mathcal{D} \leftarrow \emptyset$
3: **for** $t = 1, 2, \cdots, T - 1$ **do**
4:     Collect one step of transition $(s_t, a_t, r_t, s_{t+1})$ using the policy given by $Q_\theta$
5:     Store $(s_t, a_t, r_t, s_{t+1})$ in to replay buffer $\mathcal{D}$
6:     Update the DP value on state $s_t$,          $\triangleright$ Prioritized sweeping (Moore and Atkeson, 1993)

$$V^{DP}(s_t) \leftarrow \max_{a:(s_t,a)\in\mathcal{D}} \frac{1}{|\mathcal{D}(s_t,a)|} \sum_{(\hat{r},\hat{s}')\in\mathcal{D}(s_t,a)} \left(\hat{r} + \gamma V^{DP}(\hat{s}')\right)$$

7:     Perform one step of gradient update for $\theta$,                          $\triangleright$ Estimated by mini-batch

$$\theta \leftarrow \theta - \alpha \nabla \mathop{\mathbb{E}}_{(s,a,r,s')\sim\mathcal{D}} \left[ \left( r + \gamma \max\left\{ \max_{a'\in\mathcal{A}} Q_{\theta^{\text{target}}}(s', a'), V^{\text{DP}}(s') \right\} - Q_\theta(s, a) \right)^2 \right],$$

where $\alpha$ denotes the learning rate
8:     **if** $t \bmod H = 0$ **then**                                      $\triangleright$ Update target values
9:         $\theta^{\text{target}} \leftarrow \theta$
10:         Update the DP values over the whole replay buffer $\mathcal{D}$
11: **return** $\theta$

---

## B.4 Implementation Details and Hyper-Parameters

Our experiment environments are based on the standard Atari benchmark tasks supported by OpenAI Gym (Brockman et al., 2016). All baselines and our approaches are implemented using the same set of hyper-parameters suggested by Castro et al. (2018). More specifically, all algorithm investigated in this paper use the same set of training configurations.

- Number of `noop` actions while starting a new episode: 30;
- Number of stacked frames in observations: 4;
- Scale of rewards: clipping to $[-1, 1]$;
- Buffer size: $10^6$;
- Batch size: 32;
- Start training: after collecting 20000 transitions;
- Training frequency: 4 timesteps;
- Target updating frequency: 8000 timesteps;
- $\epsilon$ decaying: from 1.0 to 0.01 in the first 250000 timesteps;
- Optimizer: Adam with $\varepsilon = 1.5 \cdot 10^{-4}$;
- Learning rate: $0.625 \cdot 10^{-4}$.

For ensemble-based methods, Averaged DQN and Maxmin DQN, we adopt 2 ensemble instances to ensure all architectures presented in this paper use comparable number of trainable parameters.

All networks are trained using a single GPU and a single CPU core.

- GPU: GeForce GTX 1080 Ti
- CPU: Intel(R) Xeon(R) CPU E5-2630 v4 @ 2.20GHz

In each run of experiment, 10M steps of training can be completed within 36 hours.

### B.5 Abstracted Dynamic Programming for Atari Games

**State Aggregation.** We consider a simple discretization to construct the state abstraction function $\phi(\cdot)$ used in Eq. (6). We first follow the standard Atari pre-processing proposed by Mnih et al. (2015) to rescale each RGB frame to an $84 \times 84$ luminance map, and the observation is constructed as a stack of 4 recent luminance maps. We round the each pixel to 256 possible integer intensities and use a standard static hashing, Rabin-Karp Rolling Hashing (Karp and Rabin, 1987), to set up the table for storing $V_{\text{DP}}$. In the hash function, we use two large prime numbers ($\approx 10^9$) and select their primary roots as the rolling basis. From this perspective, each image would be randomly projected to an integer within a range ($\approx 10^{18}$).

**Conservative Action Pruning.** To obtain a conservative value estimation, we follow the suggestions given by Fujimoto et al. (2019) and Liu et al. (2020) to prune the unseen state-action pairs in the abstracted MDP. Formally, in the dynamic programming module, we only allow the agent to perform state-action pairs that have been collected at least once in the experience buffer. The reward and transition functions of remaining state-action pairs are estimated through the average of collected samples.

**Computation Acceleration.** Note that the size of the abstracted MDP is growing as the exploration. Regarding computational considerations, we adopt the idea of *prioritized sweeping* (Moore and Atkeson, 1993) to accelerate the computation of tabular dynamic programming. In addition to periodically applying the complete Bellman operator, we perform extra updates on the most recent visited states, which would reduce the total number of operations to obtain an acceptable estimation. Formally, our dynamic programming module contains two branches of updates:

1. After collecting each complete trajectory, we perform a series of updates along the collected trajectory. In the context of *prioritized sweeping*, we assign the highest priorities to the most recent visited states.

2. At each iteration of target network switching, we perform one iteration of value iteration to update the whole graph.

**Connecting to Parameterized Estimator.** Finally, the results of abstracted dynamic programming would be delivered to the deep Q-learning as Eq. (7). Note that the constructed doubly bounded target value $y_{\theta_{\text{target}}}^{\text{DB}}$ is only used to update the parameterized value function $Q_\theta$ and would not affect the computation in the abstracted MDP.

## C Visualization of Non-Optimal Approximate Fixed Points

As shown in Figure 4, the sample efficiency of our methods dramatically outperform double DQN in an Atari game called Zaxxon. We visualize a screen snapshot of a scenario that the double DQN agent gets stuck in for a long time (see Figure 7), which may refer to a practical example of non-optimal fixed points.

In this scenario, the agent (i.e., the aircraft at the left-bottom corner) needs to pass through a gate (i.e., the black cavity in the center of the screen). Otherwise it will hit the wall and crash. The double DQN usually gets stuck in a policy that cannot pass through this gate, but our method can find this solution very fast. We provide a possible interpretation here. In this case, the agent may find the path to pass the gate by random exploration, but when it passes the gate, it will be easier to be attacked by enemies. Due to the lack of data, the

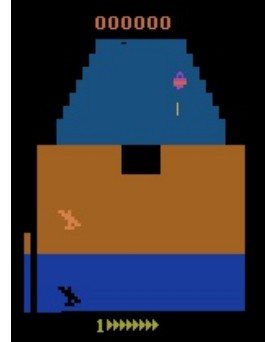

Figure 7: Visualization of a non-optimal approximate fixed point in Zaxxon.

value estimation may suffer from a large amount of error or variance in these states. It would make the agent be confused on the optimal action. By contrast, the dynamic programming planner used by double bounded Q-learning is non-parametric that can correctly reinforce the best experience.

## D  Additional Experiments on Random MDPs

In addition to the two-state toy MDP, we evaluate our method on a benchmark task of random MDPs.

**Experiment Setting.**  Following Jiang et al. (2015), we generate 1000 random MDPs with 10 states and 5 actions from a distribution. For each state-action pair $(s, a)$, the transition function is determined by randomly choosing 5 non-zero entries, filling these 5 entries with values uniformly drawn from $[0, 1]$, and finally normalizing it to $P(\cdot|s, a)$. i.e.,

$$P(s'|s, a) = \frac{\widehat{P}(s'|s, a)}{\sum_{s'' \in \mathcal{S}} \widehat{P}s''|s, a)} \quad \text{where} \quad \widehat{P}(s'|s, a) \sim \text{Uniform}(0, 1).$$

The reward values $R(s, a)$ are independently and uniformly drawn from $[0, 1]$. We consider the discount factor $\gamma = 0.99$ for all MDPs. Regarding the simulation of approximation error, we consider the same setting as the experiments in section 5.1. We consider Gaussian noise $\mathcal{N}(0, 0.5)$ and use soft updates with $\alpha = 10^{-2}$.

**Approximate Dynamic Programming.**  Our proposed method uses a dynamic programming module to construct a second value estimator as a lower bound estimation. In this experiment, we consider the dynamic programming is done in an approximate MDP model, where the transition function of each state-action pair $(s, a)$ is estimated by $K$ samples. We aim to investigate the dependency of our performance on the quality of MDP model approximation.

**Experiment Results.**  We evaluate two metrics on 1000 random generated MDPs: (1) the value estimation error $(V - V^*)$, and (2) the performance of the learned policy $(V^\pi - V^*)$ where $\pi = \arg\max Q$. All amounts are evaluated after 50000 iterations. The experiment results are shown as follows:

| Evaluation Metric | Estimation Error $(V - V^*)$ | Policy Performance $(V^\pi - V^*)$ |
| --- | --- | --- |
| double Q-learning | -16.36 | -0.68 |
| Q-learning | 33.15 | -0.73 |
| ours $(K = 10)$ | 0.53 | -0.62 |
| ours $(K = 20)$ | 0.54 | -0.62 |
| ours $(K = 30)$ | 0.37 | -0.61 |

Table 1: Evaluation on random MDPs.

We conclude the experiment results in Table 1 by two points:

1. As shown in the above table, Q-learning would significantly overestimate the values, and double Q-learning would significantly underestimate the values. Comparing to baseline algorithms, the value estimation of our proposed method is much more reliable. Note that our method only cuts off low-value noises, which may lead to a trend of overestimation. This overestimation would not propagate and accumulate during learning, since the first estimator $y^{\text{Boots}}$ has incentive to underestimate values. The accumulation of overestimation errors cannot exceed the bound of $y^{\text{DP}}$ too much. As shown in experiments, the overestimation error would be manageable.

2. The experiments show that, although the quality of value estimation of Q-learning and double Q-learning may suffer from significant errors, they can actually produce polices with acceptable performance. This is because the transition graphs of random MDPs are strongly connected, which induce a dense set of near-optimal polices. When the tasks have branch structures, the quality of value estimation would have a strong impact on the decision making in practice.

# E Additional Experiments for Variance Reduction on Target Values

Table 2: Evaluating the standard deviation of target values w.r.t. different sequences of training batches. The presented amounts are normalized by the value scale of corresponding runs. "†" refers to ablation studies where we train the network using our proposed approach but evaluate the target values computed by bootstrapping estimators.

| TASK NAME | DB-ADP-C | DB-ADP-C$^\dagger$ | CDDQN | DB-ADP | DB-ADP$^\dagger$ | DDQN |
|---|---|---|---|---|---|---|
| ALIEN | **0.006** | 0.008 | 0.010 | 0.008 | 0.009 | 0.012 |
| BANKHEIST | **0.009** | 0.010 | 0.010 | **0.009** | 0.011 | 0.013 |
| BATTLEZONE | **0.012** | **0.012** | 0.031 | 0.014 | 0.014 | 0.036 |
| FROSTBITE | **0.006** | 0.007 | 0.012 | 0.007 | 0.007 | 0.015 |
| JAMESBOND | 0.010 | 0.010 | **0.009** | 0.010 | 0.011 | 0.010 |
| MSPACMAN | **0.007** | 0.009 | 0.012 | **0.007** | 0.009 | 0.013 |
| QBERT | **0.008** | 0.010 | 0.011 | 0.009 | 0.011 | 0.012 |
| ROADRUNNER | **0.009** | 0.010 | 0.010 | **0.009** | 0.011 | 0.012 |
| STARGUNNER | **0.009** | 0.010 | **0.009** | **0.009** | 0.010 | 0.010 |
| TIMEPILOT | 0.012 | 0.013 | 0.012 | **0.011** | 0.013 | **0.011** |
| WIZARDOFWOR | **0.013** | 0.017 | 0.029 | 0.017 | 0.018 | 0.034 |
| ZAXXON | **0.010** | 0.012 | **0.010** | 0.011 | 0.011 | **0.010** |

As shown in Table 2, our proposed method can achieve the lowest variance on target value estimation in most environments.

# F Additional Experiments for Baseline Comparisons and Ablation Studies

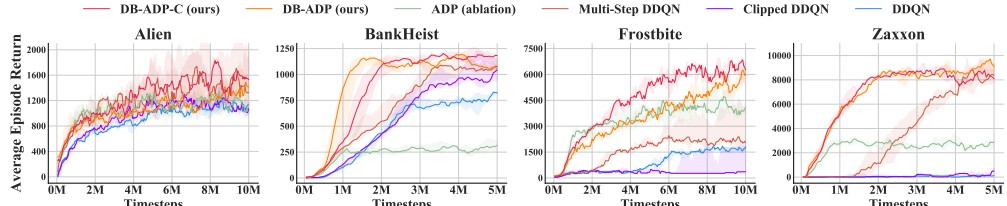

Figure 8: Learning curves on a suite of Atari benchmark tasks for comparing two additional baselines.

We compare our proposed method with two additional baselines:

- **ADP (ablation):** We conduct an ablation study that removes Bellman error minimization from the training and directly optimize the following objective:

$$L^{\text{ADP}}(\theta) = \mathop{\mathbb{E}}_{(s_t,a_t)\sim D} \left( Q_\theta(s_t, a_t) - y^{\text{DP}}(s_t, a_t) \right)^2,$$

  where $y^{\text{DP}}(s_t, a_t)$ denotes the target value directly generated by dynamic programming. As shown in Figure 8, without integrating with Bellman operator, the abstracted dynamic programming itself cannot find a good policy. It remarks that, in our proposed framework, two basis estimators are supplementary to each other.

- **Multi-Step DDQN:** We also compare our method to a classical technique named multi-step bootstrapping that modifies the objective function as follows:

$$L^{\text{Multi-Step}}(\theta; \theta_{\text{target}}, K) = \mathop{\mathbb{E}}_{(s_t,a_t)\sim D} \left( Q_\theta(s_t, a_t) - \left( \sum_{u=0}^{K-1} r_{t+u} + \gamma^K V_{\theta_{\text{target}}}(s_{t+K}) \right) \right)^2,$$

  where we select $K = 3$ as suggested by (Castro et al., 2018). As shown in Figure 8, our proposed approach also outperforms this baseline.