# OpenReview forum: "On the Estimation Bias in Double Q-Learning"
_NeurIPS.cc/2021/Conference — NeurIPS 2021 Poster_

### Official Review · Reviewer_gAo6 · 2021-07-10

**Rating:** 4
**Confidence:** 5

**Summary:**

This paper studies the effect of the underestimation induced by Double Q-Learning, a well-known method to solve the overestimation problem of Q-Learning. The paper shows that Double Q-Learning might incur in suboptimal fixed points under an approximated setting, e.g. Double DQN, and proposes a method, i.e, Doubly Bounded Q-Learning, to curb this problem. The method consists of computing the target of the Q-Learning update as a maximum operator between the application of the Bellman operator on an approximation of the value function, and the application of the Bellman operator on the optimal value function computed on a discrete abstraction of the original MDP. The method is evaluated on several Atari games.

**Limitations And Societal Impact:**

Not applicable.

**Main Review:**

The paper investigates the important problem of biased estimate of the target value of the Q-Learning update. In particular, it describes an issue of Double Q-Learning that is not documented in literature, that causes the algorithm to incur in suboptimal fixed points in approximated settings. I think the paper conducts a reasonable analysis of the problem, and supports its claims with a simple, but meaningful, toy problem.
However, despite a satisfying analysis of the problem, I consider the proposed solution not rigorous enough. Although the authors make some claims about the effectiveness of the method to escape suboptimal fixed points, these claims are not supported by theoretical reasons. The sketch of the analysis on the saddle point-nature of these fixed points, and the connection with perturbed gradient descent, are interesting considerations that I think deserve a deeper analysis. The absence of theoretical guarantees on the bias obtained by Doubly Bounded Q-Learning is an important weakness of this paper, as well as the absence of theoretical guarantees about the performance w.r.t. the quality of the approximation of the MDP model.
The empirical analysis on Atari games is not satisfying enough, as many results of the baseline are significantly different from the ones reported in literature [1,2]. In general, I am aware Atari experiments are extremely sensitive to the choice of hyperparameters, thus, I’d rather see more reliable benchmarks to complement them. A toy problem to study more rigorously the effect of Doubly Bounded Q-Learning would be desirable. Also, it would be nice to have some evidence of the estimate of the Q-function computed by Doubly Bounded Q-Learning w.r.t. other methods. I encourage authors submitting a revised version addressing these points. Overall, this work is promising, but it needs still much work to be recommendable for publication.

[1] Van Hasselt, Hado, Arthur Guez, and David Silver. "Deep reinforcement learning with double q-learning." Proceedings of the AAAI conference on artificial intelligence. Vol. 30. No. 1. 2016.
[2] Wang, Ziyu, et al. "Dueling network architectures for deep reinforcement learning." International conference on machine learning. PMLR, 2016.

**Time Spent Reviewing:**

4

---

> ### Author Response · Authors · 2021-08-10
> **Response to Reviewer gAo6**
>
> Thanks for your comments. We follow the reviewer's suggestion and conduct additional experiments to characterize the algorithmic properties of the proposed method. If our response does not fully address your concerns, please post additional questions and we will be happy to have further discussions.
>
> **Q1: Clarification on Atari Experiments**
>
> We do not agree our implementation has a significant gap to the literature.
>
> With limited computation resources, we cannot reproduce the full learning curves of DQN-based baselines that include 200M frames for each agent. That is why the reported scores seem to have a gap to the references given by the reviewer. Using the same number of training samples, our implementation of DQN is comparable to the learning curves released by Google Dopamine [1] (i.e., our 10M timesteps = 40 iterations of Dopamine).
>
> To establish a fair comparison, we use the same set of hyper-parameters in the implementations of baselines and our proposed approach. Our performance significantly outperforms baselines as shown in the experiments, which demonstrates the strength of our method.
>
> In addition, following the reviewer's suggestion, we conduct additional experiments to enrich the discussions on our proposed approach (see the following questions).
>
> *Reference*
>
> [1] Google Dopamine. https://google.github.io/dopamine/baselines/plots.html
>
> **Q2: A toy problem to study the algorithmic effects more rigorously.**
>
> As mentioned in section 3.3, our proposed method is motivated by the literature on escaping saddle points. We argue that, similar to the properties of saddle points, the non-optimal fixed points are not strongly static solutions, which can be escaped by adopting some signals with weak guidance towards the correct solution. Our proposed method utilizes a second value estimator to provide such guidance.
>
> To support these claims, we conduct an additional experiment on the two-state MDP example presented in Figure 1(a). In this example, double Q-learning may get stuck in non-optimal stationary solutions with $V(s_0)\approx 100$, and the optimal solution has $V^*(s_0)=110$. Formally, in the evaluation, we focus on the value of $V(s_0)$ since it indicates whether the agent gets stuck in a non-optimal fixed point. We consider the indicator $\mathbb{I}[V(s_0)<103]$ as the evaluation metric, i.e., $\mathbb{I}[V(s_0)<103]=\text{True}$ means the agent gets stuck. We evaluate the probability of getting stuck in the non-optimal fixed points by counting the indicator $\mathbb{I}[V(s_0)<103]$ over 1000 runs. We investigate the performance of our method with an inaccurate dynamic programming module, in which $V_{\text{DP}}(s_0)$ is set to $\\{99, 99.5, 100, 100.5, 101\\}$. The experiment results are shown as follows:
>
> | Iterations | 1000 | 3000 | 5000 | 10000 | 20000 |
> | ------ |  ------ |  ------ |  ------ |  ------ | ------ |
> | double Q-learning | 94.4\% | 84.9\% | 74.7\% | 56.7\% | 34.3\% |
> | ours ($V_{\text{DP}}(s_0)=99$) | 94.0\% | 79.3\% | 65.8\% | 45.7\% | 23.2\% |
> | ours ($V_{\text{DP}}(s_0)=99.5$) | 87.7\% | 66.2\% | 48.7\% | 24.2\% | 8.3\% |
> | ours ($V_{\text{DP}}(s_0)=100$) | 71.2\% | 31.5\% | 13.8\% | 3.4\% | 1.3\% |
> | ours ($V_{\text{DP}}(s_0)=100.5$) | 28.6\% | 3.5\% | 0.8\% | 0.4\% | 0.6\% |
> | ours ($V_{\text{DP}}(s_0)=101$) | 7.9\% | 0.4\% | 0.3\% | 0.3\% | 0.3\% |
>
> The above experiments can support our claims:
>
> - The properties of non-optimal fixed points are similar to that of saddle points. These extra fixed points are relatively stationary regions but not truly static. A series of lucky noises can help the agent to escape from non-optimal stationary regions. This procedure may take lots of iterations. For example, one-third of double Q-learning agents cannot figure out $V(s_0)>103$ even after 20000 iterations.
>
> - The design of our doubly bounded estimator is similar to a perturbation on value learning. As shown in experiments, when the estimation given by dynamic programming is slightly higher than the non-optimal fixed points, such as $V_{\text{DP}}(s_0)=100.5$, it is sufficient to help the agent escape from non-optimal stationary solutions, i.e., a weak guidance is sufficient.
>
> **Q3\&Q4: The estimate of the Q-function w.r.t. other methods. The quality of the approximation of the MDP model.**
>
> In addition to the two-state MDP experiment discussed above, we establish another experiment on a benchmark of random MDPs. We generate 1000 random MDPs with 10 states and 5 actions from a distribution. A detailed description of the experiment setting is included in the general response to all reviewers. To investigate the dependency of our performance on the quality of MDP model approximation, we consider the transition function of each state-action pair $(s,a)$ is estimated by $K$ samples.
>
> We evaluate the value estimation error $(V(s)-V^*(s))$ over baselines and our method. The experiment results are shown as follows:
>
> | Evaluation Metric | Estimation Error $(V-V^*)$ |
> | ------ |  :------: |
> | double Q-learning | -23.39 |
> | Q-learning | 85.82 |
> | ours ($K=10$) | 1.11 |
> | ours ($K=20$) | 0.81 |
> | ours ($K=30$) | 0.83 |
>
> As shown in the above table, Q-learning would significantly overestimate the values, and double Q-learning would significantly underestimate the values. Comparing to baseline algorithms, the value estimation of our proposed method is much more reliable, and the quality of value estimation is not sensitive to the quality of MDP model. Note that our method only cuts off low-value noises, which may lead to slight overestimation. This overestimation would not propagate and accumulate during learning, since the first estimator $y^{\text{Boots}}$ has an incentive to underestimate values. The accumulation of overestimation errors cannot exceed the bound of $y^{\text{DP}}$ too much. As shown in experiments, the overestimation error would be manageable.

---

> > ### Comment · Reviewer_gAo6 · 2021-08-19
> > **Comments to author response**
> >
> > I thank the authors for their clarification about the Atari experiment. I also appreciate the additional empirical results, but I still feel the paper needs a stronger theoretical study to support its claims. I'll therefore stick to my score.

---

### Official Review · Reviewer_cSd4 · 2021-07-16

**Rating:** 6
**Confidence:** 4

**Summary:**

This paper tries to address the possible underestimation issue of double Q-learning. Specifically, it shows underestimation bias may lead to multiple non-optimal fixed points, so it adds another target value from abstracted MDP to construct an integrated estimator, named doubly bounded estimator. It conducts serval experiments based on the standard Atari benchmark tasks, and the proposed doubly bounded estimator has shown promising results in bootstrapping the performance of double Q-learning algorithms.

**Main Review:**

This paper propose a simple but effective approach as a partial fix for the underestimation bias in double Q-learning. In pecular, it presents a doubly bounded estimator in Eq. 7, which takes maximum over two different bias estimators. The main idea is to leverage an abstracted dynamic programming as a second value estimator to rule out non-optimal fixed points.

Over all quality and clarity: its idea is clear, which is motived by the possible underestimation bias from double q-learning. This paper analyze non-optimal fixed points issue in both theoretically and empirically. And its experiment with a second value estimator does improve performance on a variety of Avari tasks.

Originality/Significance: this work introduces a novel abstracted dynamic programming as a second value estimator to overcome the possible underestimation bias in double q-learning.


Some issues:
How do you guarantee y^{DP} >= y^{Boots} in Eq.7 while you use abstracted MDP to get y^{DP}? Otherwise, it falls back to original double q-learning.  It is better to add the condition in  Lemma 4 to Proposition 4 in line 274. The variance result in Proposition 4 needs V^{DP} <=E[T^{Boots}] to make it holds. In addition, it seems it is not easy to satisfy both y^{DP} >= y^{Boots} and y^{DP} <= E(y^{Boots})?

Possible lack of citation:
Deep Reinforcement Learning With Adaptive Combined Critics, 2020
Decorrelated double q-learning, 2020



**Time Spent Reviewing:**

20

---

> ### Author Response · Authors · 2021-08-10
> **Response to Reviewer cSd4**
>
> Thanks for the comments. We provide clarification to your questions and concerns as below. If our response does not fully address your concerns, please post additional questions and we will be happy to have further discussions.
>
> **Q1: How do you guarantee $y^{\text{DP}} \geq y^{\text{Boots}}$?**
>
> We would like to argue three points regarding this question:
>
> - Our proposed method does not require the condition $y^{\text{DP}} \geq y^{\text{Boots}}$ always holds during learning. The main purpose of introducing $y^{\text{DP}}$ as a lower bound estimation is to help the agent escape from non-optimal fixed points. When the agent does not get stuck in non-optimal fixed points, we do not need to care whether $y^{\text{DP}} \geq y^{\text{Boots}}$ holds or not, since the Q-learning estimator can push forward the learning procedure by itself.
>
> - In non-optimal fixed points, the value estimation of double Q-learning is clearly in an underestimated status. As revealed by Proposition 3, each non-optimal fixed point corresponds to the value of a non-optimal stochastic policy. If the dynamic programming module $y^{\text{DP}}$ can approximately find the best deterministic policy within the support of the dataset, it should provide a higher estimation than $y^{\text{Boots}}$.
>
> - To achieve high learning efficiency, the value of $y^{\text{DP}}$ only needs to be slightly larger than the value of non-optimal fixed points. We conduct an additional experiment on the two-state MDP example presented in Figure 1(a). In this example, double Q-learning may get stuck in non-optimal stationary solutions with $V(s_0)\approx 100$, and the optimal solution has $V^*(s_0)=110$. Our experiments show that, to help the agent escape from this non-optimal stationary region, the second estimator only needs to output a value estimation slightly higher than the non-optimal fixed points, e.g., $V_{\text{DP}}(s_0)=100.5$. It indicates that, if the second estimator can provide some weak guidance towards the true solution, it is sufficient to help the Q-learning estimator escape from non-optimal fixed points. We include a detailed description of the experiment setting and results in the general response to all reviewers.
>
> **Q2: It is better to add the condition in Lemma 4 to Proposition 4 in line 274.**
>
> Proposition 4 does not require the condition in Lemma 4.
>
> Lemma 4 is an intermediate result which require the condition $V^{\text{DP}}\leq \mathbb{E}[V^{\text{Boots}}]$. In the proof of Proposition 4, we can apply Lemma 4 several times to remove the requirement of this condition. When $\mathbb{E}[V^{\text{Boots}}]$ is smaller than $V^{\text{DP}}$, we can first apply a lower bound cut-off by value $y=\mathbb{E}[V^{\text{Boots}}]<V^{\text{DP}}$, which gives $\text{Var}[\max\\{y,V^{\text{Boots}}\\}]\leq \text{Var}[V^{\text{Boots}}]$ and $\mathbb{E}[\max\\{y,V^{\text{Boots}}\\}]> \mathbb{E}[V^{\text{Boots}}]$. By repeating this procedure several times, we can finally get $V^{\text{DP}}\leq \mathbb{E}[\max\\{y,V^{\text{Boots}}\\}]$ and close the proof.
>
> We thank the reviewer for the careful check, and we will enrich our proof for Proposition 4 to include more details.
>
> **Q3: Possible lack of citation**
>
> We thank the reviewer for providing related papers. We will discuss these related work in the next revision.

---

> > ### Comment · Reviewer_cSd4 · 2021-09-01
> > **Thanks for your response**
> >
> > I think the authors have addressed my comments. So I will keep my rating.

---

> > > ### Author Response · Authors · 2021-09-02
> > > **Thanks for your responsive reply**
> > >
> > > Thanks for your responsive reply and for the inspiring review that helps us to improve our work.
> > >
> > > We will incorporate these discussions and refine the presentation in the next revision.

---

### Official Review · Reviewer_U61t · 2021-07-16

**Rating:** 6
**Confidence:** 3

**Summary:**

This paper proposes a new problem of double Q-learning, that is, the multiple fixed points caused by underestimation bias. This problem may lead to the non-optimal model after value iterations. Meanwhile, the authors provide many mathematical approvals. To solve it, the authors integrate another value estimator to escape the saddle points and approve its effectiveness with experiments.

**Limitations And Societal Impact:**

This paper presents methodological work and does not have foreseeable direct societal implications.

**Main Review:**

1)	The existence of multiple approximate fixed points in double Q-learning. The examples used are so special that the existence of multiple approximate fixed points are unclear in practical games. Even if there is an approval in Proposition 2, the authors do not show the visualization of fixed points of the practical games.
2)	The authors use a well-shaped perturbation mechanism to remove non-optimal saddle points. But what is a well-shaped perturbation mechanism? Maybe, different network initialization is a good perturbation. This paper lacks the definition and discussion of perturbation.
3)	The paper lacks the Algorithm table for detailed descriptions.


**Time Spent Reviewing:**

3 hours

---

> ### Author Response · Authors · 2021-08-10
> **Response to Reviewer U61t**
>
> Thanks for the comments. We provide clarification to your questions and concerns as below. If our response does not fully address your concerns, please post additional questions and we will be happy to have further discussions.
>
> **Q1: The visualization of fixed points in the practical games.**
>
> We consider an Atari game named Zaxxon as an example. In this game, the agent needs to control an aircraft to combat with enemies. The agent will receive positive rewards while destroying any enemy aircrafts. However, when the agent intends to attack an enemy, it will increase its risk to be damaged, which leads to negative rewards. Thus the value estimation would suffer from higher errors in such aggressive actions. We find that DQN-based baselines would get stuck in conservative policies that do not try to approach any enemies and receive zero rewards in a long period of learning. This phenomenon resembles a non-optimal stationary solution. In comparison, our proposed method can help the agent to figure out the correct values of aggressive actions, which accelerates the learning.
>
> We will provide a set of screen snapshots in the next revision.
>
> **Q2: On the analogy of perturbation.**
>
> The reviewer may have a little misunderstanding on our analogy of perturbation.
>
> In our proposed method, we use a lower bounded target value $y^{\text{DB}}=\max\\{y^{\text{Boots}}, y^{\text{DP}}\\}$. The first estimator $y^{\text{Boots}}$ corresponds to the standard Q-learning component. The second estimator $y^{\text{DP}}$ is the perturbation term that can help the agent escape from non-optimal fixed points. This perturbation is generated by an independently learned value estimation so that it may not get stuck in the same non-optimal solution as the Q-learning estimator.
>
> To illustrate of the perturbation, we conduct an additional experiment on the two-state MDP example presented in Figure 1(a). In this example, double Q-learning may get stuck in non-optimal stationary solutions with $V(s_0)\approx 100$, and the optimal solution has $V^*(s_0)=110$. Formally, in the evaluation, we focus on the value of $V(s_0)$ since it indicates whether the agent gets stuck in a non-optimal fixed point. We consider the indicator $\mathbb{I}[V(s_0)<103]$ as the evaluation metric, i.e., $\mathbb{I}[V(s_0)<103]=\text{True}$ means the agent gets stuck. We evaluate the probability of getting stuck in the non-optimal fixed points by counting the indicator $\mathbb{I}[V(s_0)<103]$ over 1000 runs. We investigate the performance of our method with an inaccurate dynamic programming module, in which $V_{\text{DP}}(s_0)$ is set to $\\{99, 99.5, 100, 100.5, 101\\}$. The experiment results are shown as follows:
>
> | Iterations | 1000 | 3000 | 5000 | 10000 | 20000 |
> | ------ |  ------ |  ------ |  ------ |  ------ | ------ |
> | double Q-learning | 94.4\% | 84.9\% | 74.7\% | 56.7\% | 34.3\% |
> | ours ($V_{\text{DP}}(s_0)=99$) | 94.0\% | 79.3\% | 65.8\% | 45.7\% | 23.2\% |
> | ours ($V_{\text{DP}}(s_0)=99.5$) | 87.7\% | 66.2\% | 48.7\% | 24.2\% | 8.3\% |
> | ours ($V_{\text{DP}}(s_0)=100$) | 71.2\% | 31.5\% | 13.8\% | 3.4\% | 1.3\% |
> | ours ($V_{\text{DP}}(s_0)=100.5$) | 28.6\% | 3.5\% | 0.8\% | 0.4\% | 0.6\% |
> | ours ($V_{\text{DP}}(s_0)=101$) | 7.9\% | 0.4\% | 0.3\% | 0.3\% | 0.3\% |
>
> The above experiments can support our claims:
>
> - The properties of non-optimal fixed points are similar to that of saddle points. These extra fixed points are relatively stationary regions but not truly static. A series of lucky noises can help the agent to escape from non-optimal stationary regions. This procedure may take lots of iterations. For example, one-third of double Q-learning agents cannot figure out $V(s_0)>103$ even after 20000 iterations.
>
> - The design of our doubly bounded estimator is similar to a perturbation on value learning. As shown in experiments, when the estimation given by dynamic programming is slightly higher than the non-optimal fixed points, such as $V_{\text{DP}}(s_0)=100.5$, it is sufficient to help the agent escape from non-optimal stationary solutions, i.e., a weak guidance is sufficient.
>
> We will refine our discussion on this connection in the next revision.
>
> **Q3: An algorithm table for detailed descriptions.**
>
> In the implementation, the only difference between our method and DQN-based baselines is that we use a lower bounded target value as stated in Eq.(7). The second estimator $V_{\text{DP}}$ can be replaced by any other conservative planning algorithms. We will present an algorithm box in the appendix that includes detailed descriptions of dynamic programming and data processing.

---

> > ### Comment · Reviewer_U61t · 2021-08-15
> > **Keep rating**
> >
> > The authors have addressed my concerns. Thus, I keep my rating.

---

> > > ### Author Response · Authors · 2021-08-17
> > > **Thanks for your responsive reply**
> > >
> > > Thanks for your responsive reply and for the inspiring review that helps us to improve our work.
> > >
> > > We will incorporate these discussions into the next revision.

---

### Official Review · Reviewer_GSge · 2021-07-26

**Rating:** 7
**Confidence:** 4

**Summary:**

The estimation bias is a quite general problem in Reinforcement Learning originally examined in Q-learning and due to the fact that the Bellman operator has a max operation which causes a systematic overestimation of state-action values. Double Q-learning had first been proposed as a way of mitigating this maximization bias, at the expense however of an opposite minimization bias.
This papers tackles the estimation bias in particular of Double Q-learning by proposing a new way of analyzing the problem from the perspective of the fixed points of a stochastic Bellman operator (where the stochastic components models the approximation error in minimizing the Bellman error). The main theoretical message of the paper is that the estimation bias is the result of suboptimal fixed points of the stochastic Bellman operator. Loosely based on this observation the authors then propose a "Doubly bounded estimator" which they show has a variance reduction effect. Although they do not provide any theoretical guarantees in regard to bias  reduction,t hey then test their algorithm on a few tasks of the Atari 2600 benchmark, empirically demonstrating that their algorithm results in faster rewards maximization compared to DQN, Double DQN.

**Limitations And Societal Impact:**

1. The paper gives a good overview of the relevant literature, but is failing to cite some recent related work like the following published papers:
- Zhu, R; Rigotti, M. 2021. Self-correcting Q-learning, AAAI 2021
- D’Eramo, C.; Nuara, A.; and Restelli, M. 2016. Estimatingthe Maximum Expected Value through Gaussian Approximation. ICML 2016
- Lee, D.; Defourny, B.; and Powell, W. 2013. Bias-Corrected Q-Learning to Control Max-Operator Bias in Q-Learning. IEEE ADPRL Symposium, 93–99
It would be nice if for completeness the authors considered citing these in relation to their work.
2. The paper could benefit from some additional experiments, in particular on classical RL tasks to help tease apart the benefit of the individual components of the algorithm, in the same spirit as the ablation study at the end of the paper.

**Main Review:**

The paper provides a new interesting point of view on the estimation bias in Q-learning in terms of suboptimal fixed points of a stochastic Bellman operator. This insight might have future implication of use for the field. However, there is a little bit of disconnect between this theoretical development and the algorithm proposed by the authors, in the sense that the theoretical insights do not seem to help much in motivating the proposed Doubly bounded estimator and the new training algorithm proposed by the authors, except for motivating the analysis of variance reduction. To justify the absence of theoretical guarantees in regard to bias redction, the authors state that "In general, there is no existing approach [that] can completely eliminate the estimation bias in Q-learning
algorithm." This might be right, however this statement should be qualified by specifying that there are methods to construct estimators that are guaranteed to still reduce the minimization bias of the double estimator. For instance, the papers [Zhu, R; Rigotti, M. 2021. Self-correcting Q-learning, AAAI 2021] and [D’Eramo, C.; Nuara, A.; and Restelli, M. 2016. Estimatingthe Maximum Expected Value through Gaussian Approximation. ICML 2016] each propose such an estimator. So, even if these methods are not guaranteeing a complete elimination of the estimation bias, they can still guarantee a reduction.
The experimental part on the Atari 2600 games show impressive speed up compared to DQN and Double DQN. It would be beneficial to the readership to know what criterion the authors used in order to select these specific games from Atari.
It would also be instructive to see a few benchmarks of the use of the new learning algorithm on classical RL tasks where the tabular case is "exact" in terms of state representations, to observe the behavior of the Doubly bounded estimator in this case.

**Time Spent Reviewing:**

2

---

> ### Author Response · Authors · 2021-08-10
> **Response to Reviewer GSge**
>
> Thanks for the constructive comments and additional related work. In the next revision, we will discuss these related work and refine our discussions on the literature of estimation bias. If our response does not fully address your concerns, please post additional questions and we will be happy to have further discussions.
>
> **Q1: The theoretical insights do not seem to help much in motivating the proposed method.**
>
> As mentioned in section 3.3, our proposed method is a heuristic approach inspired by the literature on escaping saddle points. We argue that, similar to the properties of saddle points, the non-optimal fixed points are not strongly static solutions, which can be escaped by adopting some signals with weak guidance towards the correct solution. Our proposed method considers a second value estimator to provide such guidance.
>
> To support these claims, we conduct an additional experiment on the two-state MDP example presented in Figure 1(a). In this example, double Q-learning may get stuck in non-optimal stationary solutions with $V(s_0)\approx 100$, and the optimal solution has $V^*(s_0)=110$. Formally, in the evaluation, we focus on the value of $V(s_0)$ since it indicates whether the agent gets stuck in a non-optimal fixed point. We consider the indicator $\mathbb{I}[V(s_0)<103]$ as the evaluation metric, i.e., $\mathbb{I}[V(s_0)<103]=\text{True}$ means the agent gets stuck. We evaluate the probability of getting stuck in the non-optimal fixed points by counting the indicator $\mathbb{I}[V(s_0)<103]$ over 1000 runs. We investigate the performance of our method with an inaccurate dynamic programming module, in which $V_{\text{DP}}(s_0)$ is set to $\\{99, 99.5, 100, 100.5, 101\\}$. The experiment results are shown as follows:
>
> | Iterations | 1000 | 3000 | 5000 | 10000 | 20000 |
> | ------ |  ------ |  ------ |  ------ |  ------ | ------ |
> | double Q-learning | 94.4\% | 84.9\% | 74.7\% | 56.7\% | 34.3\% |
> | ours ($V_{\text{DP}}(s_0)=99$) | 94.0\% | 79.3\% | 65.8\% | 45.7\% | 23.2\% |
> | ours ($V_{\text{DP}}(s_0)=99.5$) | 87.7\% | 66.2\% | 48.7\% | 24.2\% | 8.3\% |
> | ours ($V_{\text{DP}}(s_0)=100$) | 71.2\% | 31.5\% | 13.8\% | 3.4\% | 1.3\% |
> | ours ($V_{\text{DP}}(s_0)=100.5$) | 28.6\% | 3.5\% | 0.8\% | 0.4\% | 0.6\% |
> | ours ($V_{\text{DP}}(s_0)=101$) | 7.9\% | 0.4\% | 0.3\% | 0.3\% | 0.3\% |
>
> The above experiments can support our claims:
>
> - The properties of non-optimal fixed points are similar to that of saddle points. These extra fixed points are relatively stationary regions but not truly static. A series of lucky noises can help the agent to escape from non-optimal stationary regions. This procedure may take lots of iterations. For example, one-third of double Q-learning agents cannot figure out $V(s_0)>103$ even after 20000 iterations.
>
> - The design of our doubly bounded estimator is similar to a perturbation on value learning. As shown in experiments, when the estimation given by dynamic programming is slightly higher than the non-optimal fixed points, such as $V_{\text{DP}}(s_0)=100.5$, it is sufficient to help the agent escape from non-optimal stationary solutions, i.e., a weak guidance is sufficient.
>
> We will refine our discussion on this connection in the next revision.
>
> **Q2: Instructive analyses on classical RL tasks**
>
> In addition to the two-state MDP experiment discussed above, we establish another experiment on a benchmark of random MDPs. In this experiment, we show that our proposed method can effectively reduce the underestimation bias and provide better value estimation than baseline algorithms. We generate 1000 random MDPs with 10 states and 5 actions from a distribution. A detailed description of the experiment setting is included in our general response to all reviewers.
>
> In the dynamic programming, we consider the transition function of each state-action pair $(s,a)$ is estimated by $K$ samples. We evaluate the value estimation error $(V(s)-V^*(s))$ over baselines and our method. The experiment results are shown as follows:
>
> | Evaluation Metric | Estimation Error $(V-V^*)$ |
> | ------ |  :------: |
> | double Q-learning | -23.39 |
> | Q-learning | 85.82 |
> | ours ($K=10$) | 1.11 |
> | ours ($K=20$) | 0.81 |
> | ours ($K=30$) | 0.83 |
>
> As shown in the above table, Q-learning would significantly overestimate the values, and double Q-learning would significantly underestimate the values. Comparing to baseline algorithms, the value estimation of our proposed method is much more reliable, and the quality of value estimation is not sensitive to the quality of MDP model. Note that our method only cuts off low-value noises, which may lead to slight overestimation. This overestimation would not propagate and accumulate during learning, since the first estimator $y^{\text{Boots}}$ has an incentive to underestimate values. The accumulation of overestimation errors cannot exceed the bound of $y^{\text{DP}}$ too much. As shown in experiments, the overestimation error would be manageable.

---

> > ### Comment · Reviewer_GSge · 2021-08-31
> > **Thank you for thorough reply**
> >
> > I'd like to thank the authors for exhaustively addressing my concerns. I'm even more of the opinion that they did a good job on a very deserving paper.

---

> > > ### Author Response · Authors · 2021-08-31
> > > **Thank you very much for your constructive and positive comments!**
> > >
> > > Thank you very much for your constructive and positive comments!
> > >
> > > We will incorporate new experimental results and analyses into the final version.

---

### Author Response · Authors · 2021-08-10
**Additional Experiment Results (for all reviewers)**

We thank all reviewers for the constructive suggestions and inspiring comments. To better illustrate the algorithmic functionality of our proposed method, we conduct two additional experiments and present the detailed descriptions as below.

## Experiment \#1: Two-state MDP

We first conduct an experiment on the two-state MDP presented in Figure 1(a) to illustrate the effects of our proposed method on removing non-optimal fixed points.

**Evaluation Setting.** In the evaluation, we focus on the value of $V(s_0)$ since it indicates whether the agent gets stuck in a non-optimal fixed point. Formally, the optimal solution has $V(s_0)=110$ and the potential non-optimal fixed points lie in $V(s_0)\approx 100$. We consider the indicator $\mathbb{I}[V(s_0)<103]$ as the evaluation metric, i.e., $\mathbb{I}[V(s_0)<103]=\text{True}$ means the agent gets stuck in non-optimal solutions.

**Tabular Implementation.** We compare the tabular algorithms using the updating rules stated as Eq.(4) and Eq.(7), which correspond to double Q-learning and our approach, respectively. We consider the Gaussian noise $\mathcal{N}(0,1)$ as a simulation of approximation error. We allow algorithms to use soft updates $Q^{(t+1)}\gets Q^{(t)} + \alpha(\widetilde{\mathcal{T}}Q^{(t)}-Q^{(t)})$ for reducing noises, which is more close to the practical cases. We set $\alpha=0.1$ as the default parameter.

**Experiment Results.** We evaluate the probability of getting stuck in the non-optimal fixed points by counting the indicator $\mathbb{I}[V(s_0)<103]$ over 1000 runs. We investigate the performance of our method with an inaccurate dynamic programming module, in which $V_{\text{DP}}(s_0)$ is set to $\\{99, 99.5, 100, 100.5, 101\\}$. The experiment results are shown as follows:

| Iterations | 1000 | 3000 | 5000 | 10000 | 20000 |
| ------ |  ------ |  ------ |  ------ |  ------ |  ------ |
| double Q-learning | 94.4\% | 84.9\% | 74.7\% | 56.7\% | 34.3\% |
| ours ($V_{\text{DP}}(s_0)=99$) | 94.0\% | 79.3\% | 65.8\% | 45.7\% | 23.2\% |
| ours ($V_{\text{DP}}(s_0)=99.5$) | 87.7\% | 66.2\% | 48.7\% | 24.2\% | 8.3\% |
| ours ($V_{\text{DP}}(s_0)=100$) | 71.2\% | 31.5\% | 13.8\% | 3.4\% | 1.3\% |
| ours ($V_{\text{DP}}(s_0)=100.5$) | 28.6\% | 3.5\% | 0.8\% | 0.4\% | 0.6\% |
| ours ($V_{\text{DP}}(s_0)=101$) | 7.9\% | 0.4\% | 0.3\% | 0.3\% | 0.3\% |

**Summary.** These results support our claims in the paper:

- The properties of non-optimal fixed points are similar to that of saddle points. These extra fixed points are relatively stationary regions but not truly static. A series of lucky noises can help the agent to escape from non-optimal stationary regions. This procedure may take lots of iterations. For example, the agent should approximately converge after $\frac{1}{\alpha(1-\gamma)}=1000$ iterations in the above example, but one-third of double Q-learning agents cannot figure out $V(s_0)>103$ after 20000 iterations.

- The design of our doubly bounded estimator is similar to a perturbation on value learning. As shown in experiments, when the estimation given by dynamic programming is slightly higher than the non-optimal fixed points, such as $V_{\text{DP}}(s_0)=100.5$, it is sufficient to help the agent escape from non-optimal stationary solutions.

## Experiment \#2: Random MDP Benchmark

In addition to the two-state toy MDP, we evaluate our method on a benchmark task of random MDPs.

**Experiment Setting.** Following Jiang et al. (2015), we generate 1000 random MDPs with 10 states and 5 actions from a distribution. For each state-action pair $(s,a)$, the transition function is determined by randomly choosing 5 non-zero entries, filling these 5 entries with values uniformly drawn from $[0, 1]$, and finally normalizing it to $P(\cdot|s, a)$. The reward values $R(s,a)$ are independently and uniformly drawn from $[0, 1]$. We consider the discount factor $\gamma=0.99$ for all MDPs.

**Approximate Dynamic Programming.** Our proposed method uses a dynamic programming module to construct a second value estimator as a lower bound estimation. In this experiment, we consider the dynamic programming is done in an approximate MDP model, where the transition function of each state-action pair $(s,a)$ is estimated by $K$ samples. We aim to investigate the dependency of our performance on the quality of the MDP model approximation.

**Experiment Results.** We evaluate two metrics on 1000 random generated MDPs: (1) the value estimation error $(V-V^*)$, and (2) the performance of the learned policy $(V^\pi-V^*)$ where $\pi=\arg\max Q$. The experiment results are shown as follows:

| Evaluation Metric | Estimation Error $(V-V^*)$ | Policy Performance $(V^\pi-V^*)$  |
| ------ |  :------: |  :------: |
| double Q-learning | -23.39 | -1.37 |
| Q-learning | 85.82 | -0.89 |
| ours ($K=10$) | 1.11 | -0.69 |
| ours ($K=20$) | 0.81 | -0.71 |
| ours ($K=30$) | 0.83 | -0.63 |

**Summary.** We conclude these experiment results by two points:

- As shown in the above table, Q-learning would significantly overestimate the values, and double Q-learning would significantly underestimate the values. Comparing to baseline algorithms, the value estimation of our proposed method is much more reliable. Note that our method only cuts off low-value noises, which may lead to a trend of overestimation. This overestimation would not propagate and accumulate during learning, since the first estimator $y^{\text{Boots}}$ has incentive to underestimate values. The accumulation of overestimation errors cannot exceed the bound of $y^{\text{DP}}$ too much. As shown in experiments, the overestimation error would be manageable.

- The experiments show that, although the quality of value estimation of Q-learning and double Q-learning may suffer from significant errors, they can actually produce polices with acceptable performance. This is because the transition graphs of random MDPs are strongly connected, which induce a dense set of near-optimal polices. When the tasks have branch structures, the quality of value estimation would have a strong impact on the decision making in practice.

*References*

[1] Jiang, Nan, et al. "The dependence of effective planning horizon on model accuracy." Proceedings of the 2015 International Conference on Autonomous Agents and Multiagent Systems. 2015.

---

### Decision · Program_Chairs · 2021-09-27

**Decision:**

Accept (Poster)

**Comment:**

This paper received mixed reviews, but the majority of reviewers were of the opinion the paper passes the threshold to be accepted.

The topic of the paper remains of interest to the community, and I think this paper might spark additional useful thoughts and investigations, so I'm also in favour of accepting the paper for presentation at the conference.